# Leveraging eQTLs to identify individual-level tissue of interest for a complex trait

**Arunabha Majumdar**[1,2]*, **Claudia Giambartolomei**[1], **Na Cai**[3,4], **Tanushree Haldar**[5], **Tommer Schwarz**[6], **Michael Gandal**[7], **Jonathan Flint**[8], **Bogdan Pasaniuc**[1,6]*

**1** Department of Pathology and Laboratory Medicine, David Geffen School of Medicine, University of California, Los Angeles, California, United States of America, **2** Department of Mathematics, Indian Institute of Technology Hyderabad, Kandi, Telangana, India, **3** Wellcome Sanger Institute, Wellcome genome campus, Hinxton, United Kingdom, **4** European Bioinformatics Institute (EMBL-EBI), Wellcome genome campus, Hinxton, United Kingdom, **5** Institute for Human Genetics, University of California, San Francisco, California, United States of America, **6** Bioinformatics Interdepartmental Program, University of California, Los Angeles, California, United States of America, **7** Program in Neurobehavioral Genetics, Semel Institute, David Geffen School of Medicine, University of California, Los Angeles, California, United States of America, **8** Department of Psychiatry and Biobehavioral Sciences, David Geffen School of Medicine, University of California, Los Angeles, California, United States of America

\* statgen.arunabha@gmail.com (AM); pasaniuc@ucla.edu (BP)

**Data Availability Statement:** All relevant data and/ or code are available for download from the following links: https://cran.r-project.org/web/packages/eGST/index.html https://data.

## Abstract

Genetic predisposition for complex traits often acts through multiple tissues at different time points during development. As a simple example, the genetic predisposition for obesity could be manifested either through inherited variants that control metabolism through regulation of genes expressed in the brain, or that control fat storage through dysregulation of genes expressed in adipose tissue, or both. Here we describe a statistical approach that leverages tissue-specific expression quantitative trait loci (eQTLs) corresponding to tissue-specific genes to prioritize a relevant tissue underlying the genetic predisposition of a given individual for a complex trait. Unlike existing approaches that prioritize relevant tissues for the trait in the population, our approach probabilistically quantifies the tissue-wise genetic contribution to the trait for a given individual. We hypothesize that for a subgroup of individuals the genetic contribution to the trait can be mediated primarily through a specific tissue. Through simulations using the UK Biobank, we show that our approach can predict the relevant tissue accurately and can cluster individuals according to their tissue-specific genetic architecture. We analyze body mass index (BMI) and waist to hip ratio adjusted for BMI (WHRadjBMI) in the UK Biobank to identify subgroups of individuals whose genetic predisposition act primarily through brain versus adipose tissue, and adipose versus muscle tissue, respectively. Notably, we find that these individuals have specific phenotypic features beyond BMI and WHRadjBMI that distinguish them from random individuals in the data, suggesting biological effects of tissue-specific genetic contribution for these traits.

## Author summary

A significant component of the genetic susceptibility to complex traits is mediated through genetic control of gene expression in one or multiple tissues. Several studies have

broadinstitute.org/alkesgroup/LDSCORE/LDSC_
SEG_ldscores/ https://www.ukbiobank.ac.uk
https://www.cog-genomics.org/plink2".

**Funding:** B.P. acknowledges support from National
Institutes of Health (NIH) with grant numbers
R01HG009120, R01MH115676, R01HG006399,
U01CA194393, 888 T32NS048004, T32MH073526
and T32HG002536. The funders had no role in
study design, data collection and analysis, decision
to publish, or preparation of the manuscript.

**Competing interests:** The authors have declared
that no competing interests exist.

highlighted the relevance of tissue specific biological mechanisms underlying the patho-
genesis of complex traits, and have often identified multiple tissues relevant to a given
phenotype in the population. Since existing methods only prioritize tissues for a complex
phenotype in the population, it remains an open question whether certain classes of indi-
viduals have their genetic predisposition for the phenotype mediated primarily through a
specific tissue. We present an efficient statistical approach that integrates tissue-specific
eQTLs (i.e., eQTLs for tissue-specific genes) with genetic association data for a complex
trait to probabilistically quantify the tissue-wise genetic contribution to the phenotype of
each individual in the study. Using simulations we show that the proposed approach accu-
rately infers the simulated tissue of interest for each individual. Integrating expression
data from the GTEx consortium, we apply the proposed approach to two obesity related
phenotypes in the UK Biobank. Our approach identified subgroups of individuals with
their genetic susceptibility to the phenotype mediated in a tissue-specific manner. Inter-
estingly, multiple metabolic traits, neuropsychiatric traits, and other traits were found to
be differentially distributed between the tissue-specific groups of individuals and the
remaining population, suggesting a biologically meaningful interpretation for these sub-
groups of individuals. We provide an R-package 'eGST' for general use of the method:
https://cran.r-project.org/web/packages/eGST/index.html.

## Introduction

Multiple clinical, pathologic, and molecular lines of evidence suggest that many phenotypes
and diseases show heterogeneity and can be viewed as a collection of multiple traits (i.e. sub-
types) in the population [1–5]. Traditional subtype identification has relied on detecting bio-
markers or subphenotypes that distinguish subsets of individuals in a biologically meaningful
way. For example, breast cancer has two well-known subtypes, estrogen receptor positive and
negative [6–8]. Patients with psychiatric disorders can have different severities [9]. With the
advent of large scale genome-wide association studies (GWAS) that have robustly identified
thousands of risk variants for complex traits, multiple approaches have investigated the use of
genetic risk variants to define classes of individuals that show genetic heterogeneity across sub-
types [10–15]. For example, autism can be subtyped by grouping together individuals with
recurrent mutations in the same autism-associated gene [10, 12]; type 2 diabetes can be sub-
typed using clusters of genetic variants previously associated with the disease [13]. Other
examples include adiposity traits such as body mass index (BMI), waist-to-hip ratio (WHR),
and WHR adjusted for BMI (WHRadjBMI), that can be subtyped based on genetic variants
with distinct patterns of fat depots and metabolism [14]. Genetic subtyping offers an advantage
over phenotypic subtyping in that germline genetic characteristics are more stable than pheno-
typic characteristics of an individual [12, 13].

A significant component of the genetic susceptibility of complex traits is mediated through
genetic control of gene expression in one or multiple tissues [16–18], with several studies
highlighting the relevance of tissue or cell type specific biological mechanisms underlying the
pathogenesis of complex traits [19–25]. Such studies rely on integration of expression quanti-
tative loci (eQTLs) with GWAS data in a tissue or cell type specific manner to prioritize tissues
and cell types that are relevant for a given complex trait. For example, a recent study [23] pro-
posed a novel approach based on the stratified LD score regression to identify the disease-rele-
vant tissues; it investigates whether the collection of genomic regions surrounding the set of
tissue-specific expressed genes are enriched for the disease heritability. These studies have

often identified multiple tissues relevant to a given trait in the population (e.g., liver, pancreas and thyroid tissues for total cholesterol [22]; connective skeletal muscle and adipose for WHRadjBMI [23, 26]; brain and adipose for BMI [21, 23, 27]). Transcriptome-wide association studies (TWAS) have identified novel associations between the genetic component of gene expression and a phenotype [16, 28–30]; genes significantly associated with the phenotype have often been discovered in multiple tissues which demonstrates the tissue-specific genetic contribution to the trait across different tissues.

These findings motivate us to ask an important follow-up question that, if a phenotype has two or more relevant tissues in the population how we can quantify the tissue-wise genetic contribution to the trait for a given individual. A method addressing this objective would also allow us to explore the possibility of prioritizing a specific tissue relevant for the phenotype of an individual. A recent study [31] has proposed to identify disease subtypes by integrating clinical features related to the disease and gene expression profiles across patients. Using a multi-view clustering algorithm they classified patients into subgroups where each subgroup has a distinct pattern of genetic component of gene expression predicted using genotypes of cis-SNPs and various clinical features related to the disease combined together. However, their approach does not investigate tissue-specific genetic predisposition to the phenotype for a given individual. Since, tissue-specific genetics plays an important functional role underlying the tissue-specific biological processes, we aim to explicitly quantify tissue-wise genetic contribution to a phenotype at an individual-level, and identify subgroups of individuals who are homogeneous with respect to the effect of tissue-specific genomic profile on the phenotype.

In this article, we present a statistical approach that integrates tissue-specific eQTLs (i.e., eQTLs for tissue-specific genes) with genetic association data for a complex trait to probabilistically quantify the tissue-wise genetic contribution to the phenotype of each individual in the study. Following previous studies [20, 23] we characterize a tissue by a set of genes specifically over-expressed in it, and develop our model based on the eQTLs for such genes in the tissue. We focus on traits where multiple tissues have been implicated by previous studies (e.g. brain and adipose for BMI [21, 23, 27]), and hypothesize that the genetic predisposition to the trait for a subgroup of individuals can act primarily through a specific tissue. Assuming that two tissues are relevant for the trait in the population, there are three possibilities for a given individual: the genetic susceptibility of the trait is mediated primarily through first tissue, second tissue, or both tissues. That is, for one group of individuals the genetic predisposition to BMI acts through regulation in the brain, for another group of individuals the genetic predisposition acts through adipose, and for the remaining individuals it acts through both. In our study, we focus on the first two subgroups of individuals and examine the characteristics that distinguish them from the remaining population. We propose eGST (eQTL-based Genetic Sub-Typer), an approach that estimates the posterior probability that whether a relevant tissue can be prioritized for an individual's phenotype or not, based on individual-level genotype data of tissue-specific eQTLs and marginal phenotype data. eGST implements a Bayesian framework of mixture model by employing a computationally efficient maximum a posteriori (MAP) expectation-maximization (EM) algorithm to estimate the tissue-specific posterior probabilities per individual.

We perform extensive simulations using real genotypes from the UK Biobank and show that eGST accurately infers the simulated tissue of interest for each individual. We also show that a Bayesian framework of the mixture model performs better than the corresponding frequentist framework. By integrating expression data from the GTEx consortium [17, 18], we apply eGST to two obesity related measures (BMI and WHRadjBMI) in the UK Biobank [32, 33]. We consider brain and adipose tissues for BMI to identify two subgroups, one with adipose and the other with brain tissue-specific genetic contribution (in aggregate 25192

individuals, 7.5% of total sample size). Similarly, we consider muscle and adipose tissues for WHRadjBMI to identify two subgroups with muscle- or adipose-specific genetic contribution. Interestingly, the subgroups of individuals classified into each tissue show distinct genetic and phenotypic characteristics. 85 out of 106 phenotypes tested in the UK Biobank were differentially distributed between the BMI-adipose (or BMI-brain) group of individuals and the remaining population, with 72 out of 85 remaining significant after adjusting for BMI. For example, diabetes proportion, various mental health phenotypes, alcohol intake frequency, and smoking status were differentially distributed between one or both of the tissue-specific groups of BMI and the remaining population. Overall, our results suggest that tissue-specific eQTLs can be successfully utilized to quantify tissue-wise genetic contribution to a complex trait and prioritize the tissue of interest at an individual-level in the study.

## Results

### Overview of methods

We start by depicting the main intuition underlying our hypothesis and model (Fig 1). For simplicity, consider two tissues of interest and assume that gene A is only expressed in tissue 1 whereas gene B is only expressed in tissue 2. The main hypothesis underlying our model is that the genetic susceptibility of a complex trait for a given individual is mediated through regulation of either gene A in tissue 1 or gene B in tissue 2 or both. Having gene expression measurements in every individual at both genes in both tissues can be used to test this hypothesis. Unfortunately, gene expression measurements in large sample sizes such as the UK biobank are not typically available. To circumvent this, since eQTLs explain a substantial heritability of gene expression [16, 34], we use eQTLs for each gene in the corresponding tissue as a proxy for the genetically regulated component of the expression. In details, eGST takes as input the phenotype values and the genotype values at a set of variants known to be eQTLs for tissue-specific genes (Fig 1). We consider a set of genes overexpressed in a tissue as the set of tissue-specific genes [20, 23]. Formally, the phenotype for individual $i$ under the tissue of interest $k$ is modeled as $y_i = \alpha_k + x'_{ki}\beta_k + \epsilon_{ki}$, where $\alpha_k$ is the baseline tissue-specific trait mean, $x_{ki}$ is the vector of normalized genotype values of individual $i$ at the eQTL SNPs specific to tissue $k$, $\beta_k$ are their effects on the trait, and $\epsilon_{ki}$ is a noise term, $i = 1, \ldots, n$ and $k = 1, \ldots, K$. For simplicity of exposition, we introduce indicator variables for each individual $C_i = k$ iff the individual $i$ has its tissue of interest $k$. Thus, $P(C_i = k) = w_k$ is the prior proportion of individuals for whom the phenotype has $k^{th}$ tissue-specific genetic effect. We assume that the eQTL SNP sets across $k$ tissues are non-overlapping and that each element in $\beta_k$, the genetic effect of $k^{th}$ tissue-specific eQTLs (i.e., eQTLs for tissue-specific genes) on the trait, is drawn from $N(0, \sigma^2_{x_k})$. If $\sigma^2_{y_k}$ is the variance of the trait under $C_i = k$, then $\sigma^2_{x_k} = \frac{h^2_k \sigma^2_{y_k}}{m_k}$, where $h^2_k$ is the heritability of the trait under $C_i = k$ due to $k^{th}$ tissue-specific $m_k$ eQTLs, and is termed as $k^{th}$ tissue-specific subtype heritability. Under this mixture model, the likelihood of individual $i$ takes the form:

$f(y_i|\Theta) = \sum_{k=1}^{K} f(y_i|C_i = k, \Theta)P(C_i = k) = \sum_{k=1}^{K} w_k \phi(y_i|\mu = \mu_{ik}, \sigma^2 = \sigma^2_{\epsilon_k})$, where $\phi(.|.)$ denotes the normal density, $\mu_{ik} = \alpha_k + x'_{ki}\beta_k$, $\Theta = (\theta_1, \ldots, \theta_K)$ with $\theta_k$ denoting $k^{th}$ tissue-specific set of model parameters. We propose a Bayesian inference approach based on a maximum a posteriori (MAP) expectation maximization (EM) algorithm to estimate the posterior probability that the phenotype of individual $i$ is mediated through the genetic effects of eQTLs specific to tissue $k$ ($P(C_i = k|X, Y)$). Our main inference is based on this posterior probability. For example, while $P(C_i = 1|X, Y) > 65\%$ indicates that tissue 1 is likely to be the tissue of interest for individual $i$, a value of the posterior probability around 50% indicates that the genetic susceptibility for the individual does not mediate through a specific tissue.

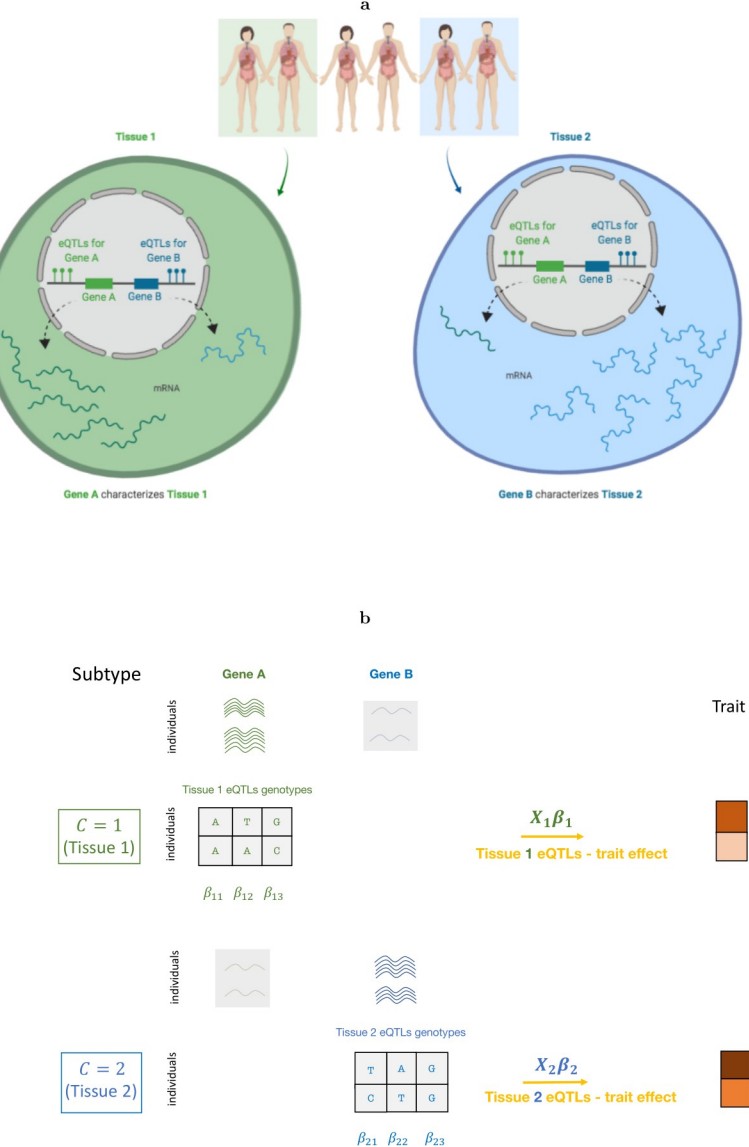

**Fig 1. The top diagram (a) explains our main hypothesis and the bottom diagram (b) explains our model.** (a): Consider two tissues of interest for a phenotype, tissue 1 and tissue 2, where gene A has higher expression but gene B has much lower expression in tissue 1, and gene B has higher expression but gene A has much lower expression in tissue 2. The key hypothesis is that the susceptibility of the phenotype for the first two individuals is mediated through the effect of gene A in tissue 1, in which case we can assign tissue 1 as the tissue of interest for these individuals (similarly tissue 2 for last two individuals). We refer to the phenotype of the first two individuals as tissue 1 specific subtype and the phenotype of last two individuals as tissue 2 specific subtype. However, two individuals in the middle are not assigned as any of the two tissue-specific subtypes and remain unclassified. (b): We use genotypes at the tissue-specific eQTLs (i.e., eQTLs for tissue-specific genes) as a proxy for the expressions of the corresponding tissue-specific genes. We consider a finite mixture model with each of its components being a linear model regressing the trait on the genotypes at each set of tissue-specific eQTLs. Our method takes as input the individual-level measurements of the phenotype and genotypes at the sets of tissue-specific eQTLs and provides per-individual tissue-specific posterior probabilities as the main output.

## Simulations

We performed simulations to assess the performance of eGST with respect to the accuracy of classifying the tissue of interest across individuals under various scenarios. We simulated

phenotypes using the real genotype data from the UK Biobank, in which two tissue-specific eQTL effects generate the phenotype (see Materials and methods). We evaluated the classification accuracy of eGST with respect to the variance explained in the trait specific to the two tissues by the two sets of tissue-specific SNPs' effects. As expected, the average area under the curve (AUC) increases with the tissue-specific subtype heritability, ranging from AUC of 50% when $h_1^2 = h_2^2 = 0\%$ to 95% when $h_1^2 = h_2^2 = 90\%$ (Fig 2). This is likely due to larger tissue-specific subtype heritabilities inducing better differentiation between the tissue-specific genetic effects. Next, we assessed the performance of eGST compared to a variation of our approach that assumes the parameters of the model to be known. The performance obtained by this gold-standard strategy can be viewed as the maximum achievable under our proposed framework. We find that eGST loses 1.4%–3.9% AUC on average compared to this strategy across all simulation scenarios considered (Fig 2 and S1 and S2 Figs). We also considered a thresholding scheme on the tissue-specific posterior probabilities to balance total discoveries versus accuracy. As expected, the true discovery rate (TDR) of classifying the tissue of interest increases (hence FDR = 1-TDR decreases) with the posterior probability threshold but the proportion of discovery decreases (Fig 3).

We then explored the effect of other parameters on the classification accuracy. First, we found that increasing sample size $n$ from 40, 000 to 100, 000 marginally increases the AUC by an average of 1% across different simulation scenarios (S2 Table), which indicates that increasing sample size improves the overall classification accuracy. Second, we observed that as the number of causal SNPs explaining a fixed heritability of each subtype increases, the average AUC marginally decreases. For example, for a fixed subtype heritability explained, the average AUC for 2000 causal SNPs (1000 per tissue) is 1% higher than that for 3000 causal SNPs (1500 per tissue) across different choices of other simulation parameters (S3 Table). Third, as the difference between the baseline tissue-specific mean of the trait across tissues increases, the classification accuracy also increases. For example, we find that the AUC increased from 60% (for no difference in tissue-specific phenotype means, $\alpha_1 = \alpha_2 = 0$) to 63% when $\alpha_1 = 0, \alpha_2 = 1$ (S4 Table). We also explored the impact of the difference between the mean of tissue-specific genetic effect size distributions and observed that the classification accuracy improves compared to zero mean of both causal effects. For example, the AUC increases from 60% to 63% if we consider $E(\beta_{1j}) = -0.02$ and $E(\beta_{2j}) = 0.02, j = 1, \ldots, 1000$, instead of zero means of $\boldsymbol{\beta}_1, \boldsymbol{\beta}_2$ (S5 Table).

Next, we explored the comparative performance of the MAP-EM algorithm under Bayesian framework (Algorithm 1) and the EM algorithm under frequentist framework (Algorithm 2). Although both approaches yield similar AUC, MAP-EM performed better than EM with respect to the true discovery rate (TDR) at different posterior probability thresholds in nearly all of the simulation scenarios (Fig 4 and S3 and S4 Figs). MAP-EM offered an average of 0.05%–20% higher TDR (hence lower FDR) than EM across various posterior probability thresholds (Fig 4 and S3 and S4 Figs).

In the above simulation scenarios, we considered that each individual has genetic contribution due to one of the tissue-specific set of SNPs. However, a group of individuals can have genetic effect due to both of the tissue-specific sets of SNPs. For such individuals, we would expect that the tissue-specific subtype posterior probability be distributed around half. We consider a sample of 30,000 individuals, in which first group of 10K individuals have genetic effect only due to the first tissue-specific set of SNPs. Second group of 10K individuals have genetic effect due to both tissue-specific sets of SNPs. And the last group of 10K individuals have effect only due to the second tissue-specific set of SNPs. We observe that the mean of the first tissue-specific subtype posterior probability for the individuals who have effect from both

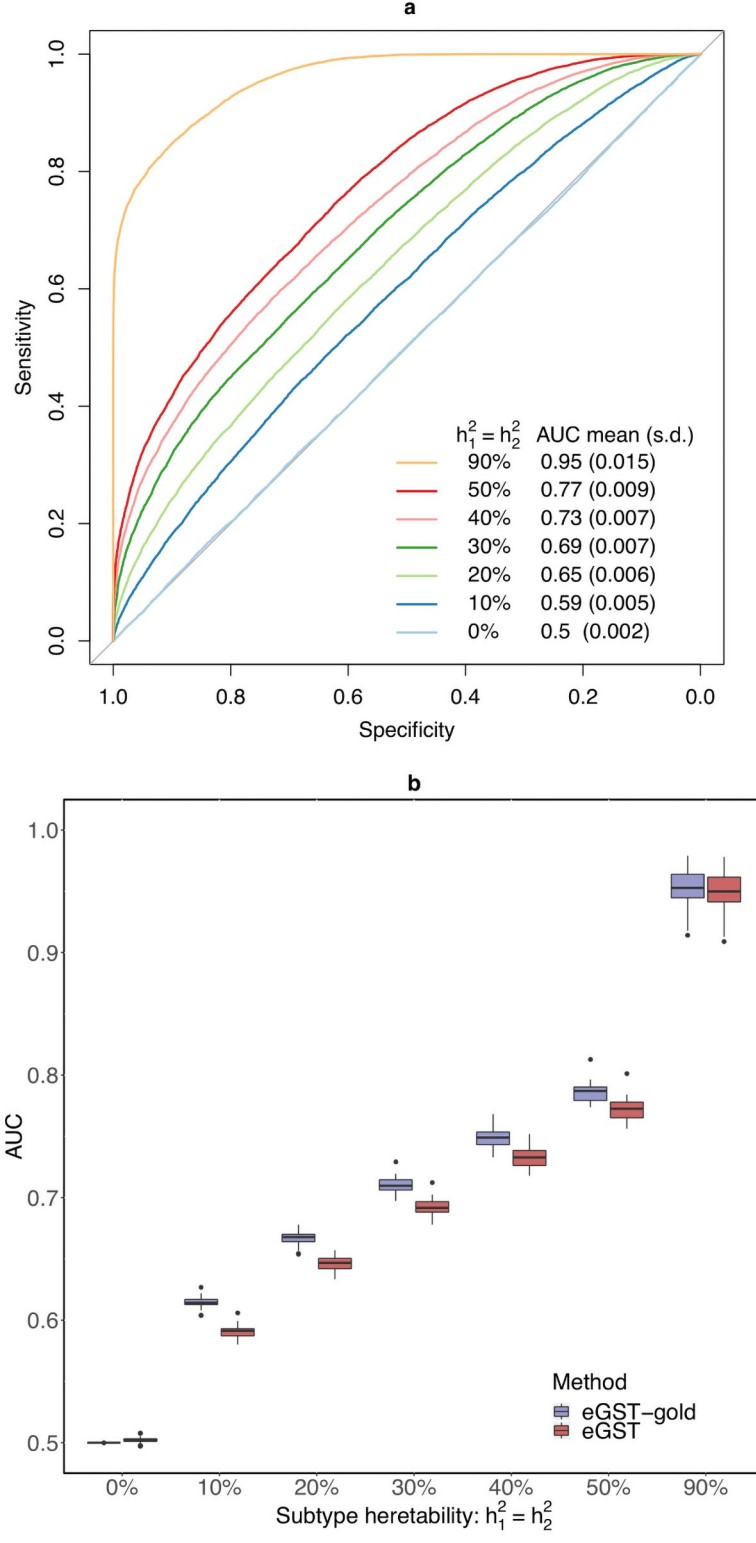

**Fig 2.** In the first diagram (a), we present the receiver operating characteristic (ROC) curve evaluating the classification accuracy of eGST for a single dataset simulated under the following scenarios: $h_1^2 = h_2^2 = 0\%, 10\%, 20\%, 30\%, 40\%, 50\%, 90\%, w_1 = w_2 = \frac{1}{2}, m_1 = m_2 = 1000, n = 40000$. The mean (across 50 simulated datasets) area under the curve (AUC) obtained by eGST under the same scenarios are also provided. Here $h_1^2$ and $h_2^2$ are the heritability of tissue-specific subtypes of the trait due to $m_1$ and $m_2$ SNPs representing two sets of tissue-

specific eQTL SNPs, $w_1$ and $w_2$ are the proportions of individuals in the sample assigned to the two tissues, $n$ is the total number of individuals. In the second diagram (b), box plots of AUCs obtained by eGST and the gold-standard strategy implementing our model in which true model parameters were assumed to be known while estimating the tissue-specific posterior probabilities are presented across the same simulation scenarios.

tissues is centered around 50% (S6 Table). As expected, the same quantity for the first group of individuals becomes larger than 50% as the tissue-specific subtype heritability increases (S6 Table).

So far, we have assumed that both of the tissues included in the simulation model are relevant for the complex trait. However, it is important to explore the performance of eGST if one of the tissues considered is completely irrelevant for the trait. We perform simulations based on a sample of 20,000 individuals where all individuals have genetic effect due to one tissue-specific set of SNPs. We run eGST including another tissue-specific set of SNPs which is irrelevant for the phenotype. We observe that a small percentage of individuals were misclassified to the irrelevant tissue, and the percentage of correctly classified individuals is much larger (S7 Table). For example, when the heritability of the trait due to the relevant tissue-specific set of SNPs is 20%, the percentage of correctly classified individuals is 89% whereas the percentage of misclassified individuals is 1.4%. The percentage of misclassification decreases as the choice of the threshold of the tissue-specific subtype posterior probability increases.

### Inferring individual-level tissue of interest for BMI and WHRadjBMI

Having established in simulations that our approach is effective in correctly classifying the individual-level tissue of interest, we next analyzed BMI and WHRadjBMI, two phenotypes that are known to have multiple tissues of interest mediating their genetic susceptibility [21, 23, 26, 27, 35–37]. For BMI analysis, we used phenotype and genotype data for 336, 106 individuals in the UK Biobank [32, 33] at 1705 adipose specific eQTLs (i.e., eQTLs for adipose-specific genes) and 1478 brain specific eQTLs (see Materials and methods). We considered rank-based inverse normal transformation of the BMI residual obtained after adjusting BMI for age, sex, and 20 PCs of genetic ancestry to adjust for population stratification. Each SNP eQTL is the top cis-association of a tissue-specific expressed gene [23] in the corresponding tissue [17, 18] (see Materials and methods). At 65% threshold of tissue-specific subtype posterior probability, 7.5% of all the individuals where assigned a tissue (adipose or brain) and the rest of the individuals remained unclassified; eGST classified the genetic susceptibility on the BMI of 11, 838 individuals through adipose eQTLs and for 13, 354 individuals through brain eQTLs (S1 Table). Individuals classified to each of the tissues are distributed across different bins of BMI (S8 Table). While the individuals classified into adipose have a higher mean of BMI (30.4) than the population, the BMI mean for brain-specific individuals (27.6) is very close to the population mean (27.4) (S10 Table).

For WHRadjBMI, we included 953 adipose subcutaneous (abbreviated and referred as AS in the following) tissue-specific eQTLs (i.e., eQTLs for AS-specific genes) and 1052 muscle skeletal connective (abbreviated as MS) tissue-specific eQTLs; and inverse normal transformed WHRadjBMI residual (adjusting WHRadjBMI for age, sex, and top 20 PCs) for 336, 018 individuals. Similarly to the BMI analysis, the tissue of interest for WHRadjBMI of a small percentage (5.7%) of all individuals were classified (S1 Table), and the remaining individuals were unclassified. Individuals assigned to the two tissues are both spread across different bins of WHRadjBMI (S9 Table). The individuals classified into MS have a higher mean of WHRadjBMI and WHR (0.03 and 0.91) than the population, and the mean for AS-specific individuals (−0.04 and 0.85) is lower than the population mean (0 and 0.87) (S11 Table).

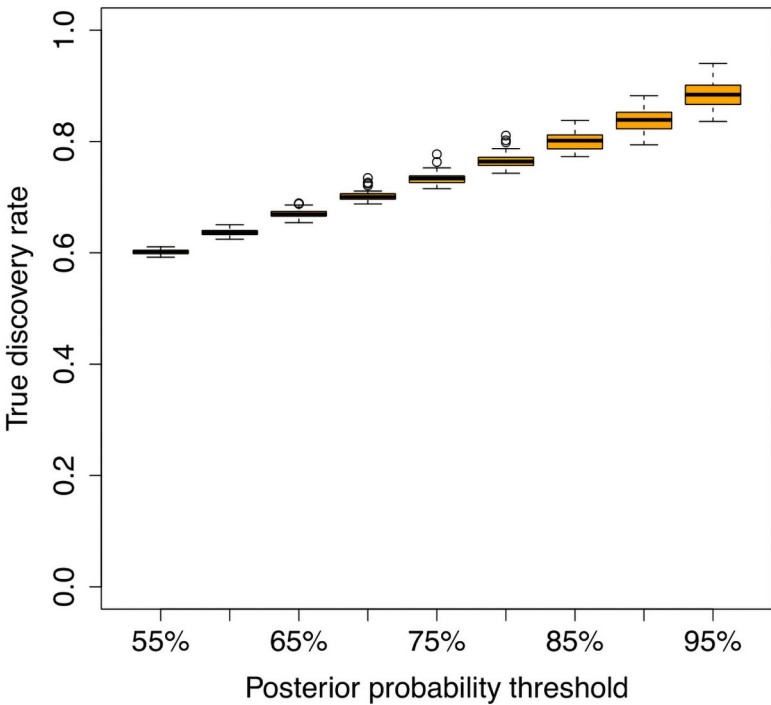

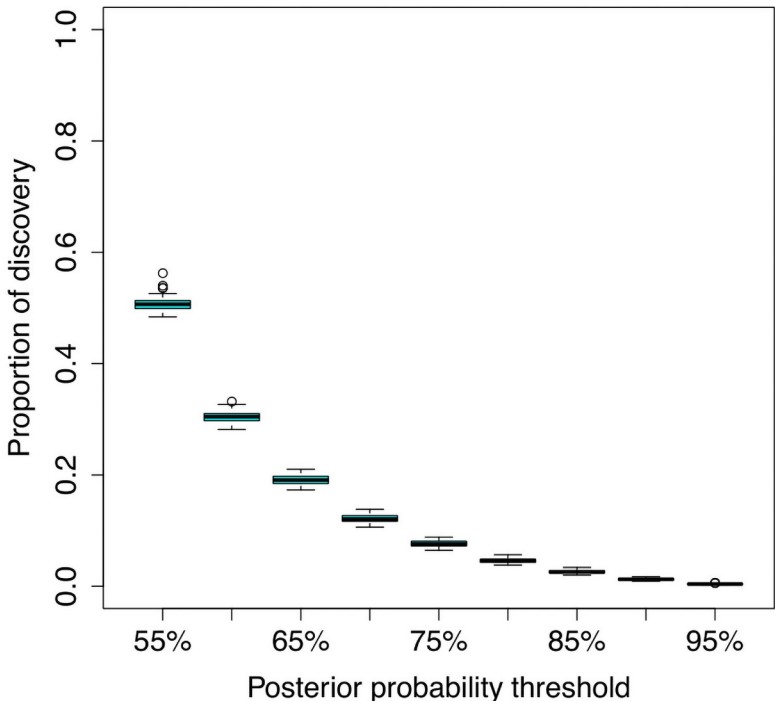

**Fig 3. True discovery rate (TDR) and the proportion of discovery (POD) by eGST while classifying the tissue of interest across individuals at increasing thresholds of tissue-specific subtype posterior probability: 55%, 60%, 65%, . . ., 90%, 95%.** Box plots of TDR (top) and POD (bottom) across 50 datasets simulated under $h_1^2 = h_2^2 = 10\%$, $w_1 = w_2 = \frac{1}{2}$, $m_1 = m_2 = 1000$, $n = 40,000$ are presented. Here $h_1^2$ and $h_2^2$ are the heritabilities of the tissue-specific subtypes of the trait due to $m_1$ and $m_2$ SNPs representing two sets of tissue-specific eQTL SNPs, $w_1$ and $w_2$ are the proportions of individuals in the sample assigned to the two tissues, $n$ is the total number of individuals.

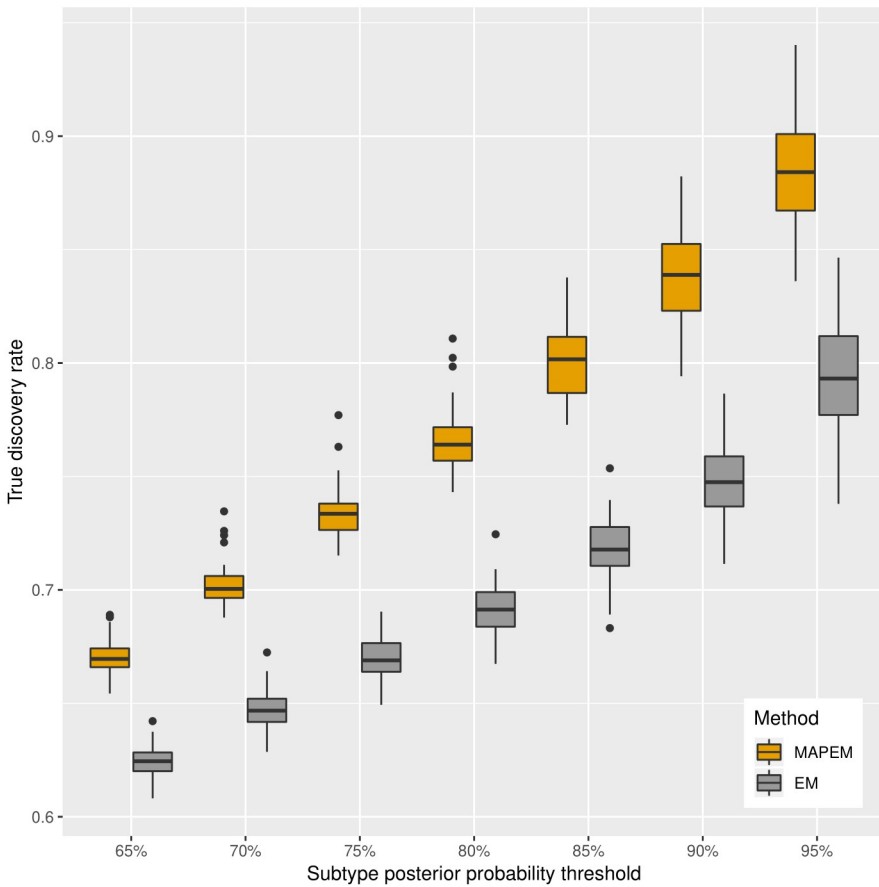

**Fig 4. Comparison between the true discovery rate (TDR) of classifying tissue-specific subtypes by the MAP-EM algorithm (under the Bayesian framework of the mixture model which eGST employs) versus the EM algorithm (under the frequentist framework of the mixture model) based on the threshold of tissue-specific subtype posterior probability as 65%, 70%, 75%, 80%, 85%, 90%, 95%, respectively.** Box plots of TDR across 50 datasets simulated under $h_1^2 = h_2^2 = 10\%$, $w_1 = w_2 = \frac{1}{2}$, $m_1 = m_2 = 1000$, $n = 40,000$ are presented. Here $h_1^2$ and $h_2^2$ are the heritability of tissue-specific subtypes of the trait due to $m_1$ and $m_2$ SNPs representing two sets of tissue-specific eQTL SNPs, $w_1$ and $w_2$ are the proportions of individuals in the sample assigned to the two tissues, $n$ is the total number of individuals.

We permuted the phenotype data across individuals while keeping the eQTL assignment to tissues fixed as it is in the original data (see Materials and methods). For BMI, the average number of individuals classified as a tissue-specific subtype (based on 65% threshold of subtype posterior probability) across 500 random permutations of the phenotype was 7404 (s.d. 700) which is substantially smaller than 25,192 individuals classified as real adipose or brain specific subtype of BMI in the original data. For WHRadjBMI, the average number of individuals classified as a tissue-specific subtype across 500 random permutations of the phenotype was 3433 (s.d. 517) compared to 19,041 individuals classified as real AS and MS specific subtypes of WHRadjBMI.

To estimate the tissue-specific subtype heritability, we employed an MCMC algorithm that implements eGST under the Bayesian framework (S1 Algorithm in S1 Text). Through simulations we found that the MCMC estimates the tissue-specific subtype heritability more unbiasedly than the MAP-EM algorithm (results not provided for brevity), but is computationally much slower. In BMI analysis, the posterior mean of the tissue-specific subtype heritability was found to be 3.6% (posterior s.d. 0.2%) and 3.7% (posterior s.d. 0.2%) for brain and adipose,

**Table 1. Genetic heterogeneity between groups of individuals assigned to tissue-specific subtypes of BMI (or WHRadjBMI).** In the BMI analysis, $\beta_1$ denotes the SNP-effect of an adipose eQTL on BMI in those individuals assigned to the adipose-specific subtype of BMI, $\beta_2$ is the SNP-effect of a brain eQTL on the brain-specific subtype of BMI, $\gamma_1$ is the SNP-effect of a brain eQTL on the adipose subtype and $\gamma_2$ is the effect of an adipose eQTL on the brain subtype. We provide the mean magnitude of the effect sizes of a tissue-specific eQTLs on the BMI of the corresponding tissue-specific group of individuals (e.g., joint SNP-effect of the adipose eQTLs on the BMI of individuals with adipose subtype), and the p-values obtained from the Wilcoxon rank sum (WRS) right tail tests of effect heterogeneity. For each test, the alternative hypotheses are listed in parentheses, while the null hypothesis is the equality between the corresponding pair of parameters. These parameters are defined in the same way for the adipose subcutaneous (AS) and muscle skeletal (MS) tissue-specific subtype of WHRadjBMI and the same analyses are performed.

| BMI analysis | | | | |
|---|---|---|---|---|
| BMI subtype | Mean magnitude of eQTLs' effects | | P-values | |
| | Adipose | Brain | | |
| Adipose | $0.045\ (|\beta_1|)$ | $0.028\ (|\gamma_1|)$ | <2E-16 $(|\beta_1| > |\gamma_1|)$ | <2E-16 $(|\beta_2| > |\gamma_1|)$ |
| Brain | $0.034\ (|\gamma_2|)$ | $0.054\ (|\beta_2|)$ | <2E-16 $(|\beta_1| > |\gamma_2|)$ | <2E-16 $(|\beta_2| > |\gamma_2|)$ |
| **WHRadjBMI analysis** | | | | |
| WHRadjBMI subtype | Mean magnitude of eQTLs' effects | | P-values | |
| | AS | MS | | |
| AS | $0.0007\ (|\beta_1|)$ | $0.0004\ (|\gamma_1|)$ | <2E-16 $(|\beta_1| > |\gamma_1|)$ | <2E-16 $(|\beta_2| > |\gamma_1|)$ |
| MS | $0.0005\ (|\gamma_2|)$ | $0.0007\ (|\beta_2|)$ | E-9 $(|\beta_1| > |\gamma_2|)$ | 9E-16 $(|\beta_2| > |\gamma_2|)$ |

respectively. Similarly, in WHRadjBMI analysis, the posterior mean of the tissue-specific subtype heritability was 3.1% (posterior s.d. 0.2%) and 2.5% (posterior s.d. 0.2%) for adipose and muscle, respectively. Inclusion of only the top eQTLs in our analyses (mainly for computational simplicity) partly explains the lower estimates of tissue-specific subtype heritability.

**Genetic characteristics.** To confirm that eGST identified groups of individuals with different genetic basis, we contrasted the SNP effects of the adipose and brain-specific eQTLs on the BMI in those individuals assigned to the adipose (or the brain specific subtype). As expected, we find that in the individuals classified as having the brain-specific subtype in their genetic contribution to BMI, the magnitude of the effect size of a brain eQTL SNP is larger than the corresponding effect size magnitude of an adipose eQTL SNP (Wilcoxon rank sum (WRS) right tail test p-value $< 2.2 \times 10^{-16}$ (Table 1)). The opposite is true for the individuals classified as having the adipose-specific subtype of BMI. We also find that the magnitude of effect size of the adipose eQTLs are larger in the adipose-specific individuals than that in the brain-specific individuals, and we find statistical evidence supporting the analogous hypothesis about the brain eQTLs, brain-specific, and adipose-specific individuals of BMI (Table 1). We observe the same pattern in our analogous analysis for WHRadjBMI (Table 1).

**Phenotypic characteristics of individuals with a prioritized tissue.** Next, we explored the phenotypic characteristics of the individuals assigned with a prioritized tissue. We considered 106 phenotypes in the UK Biobank and tested each one for being differentially distributed (heterogeneous) between the individuals of each tissue-specific subtype and the remaining population (see Materials and methods). In aggregate for BMI, 45 quantitative traits and 40 qualitative traits (total 85 among 106) were significantly heterogeneous between at least one of the BMI-adipose or BMI-brain specific groups versus the remaining population (S10 and S12 Tables and Table 2). None of these 106 traits was found to be differentially distributed between a random set of individuals from the population (with the same size as a tissue-specific subtype group) and the remaining population (see Materials and methods). 33 quantitative and 34 categorical traits showed heterogeneity in both the adipose group versus the population, and the brain group versus population. We found 6 quantitative and 3 categorical traits heterogeneous for individuals in the adipose group but not the brain group, and found 6 quantitative and 3 categorical traits heterogeneous for individuals in the brain group but not the adipose group (S10 and S12 Tables and Table 2).

**Table 2. Qualitative/Categorical traits with three or more categories that are differentially distributed between at least one of the adipose and brain-specific subtype groups of individuals for BMI and the remaining population.** For each trait, we provide the p-values of testing heterogeneity between each tissue-specific subtype group of individuals and the remaining population before (primary) and after BMI adjustment (BMIadj). For each trait, tissue-specific groups which appear to be significantly heterogeneous (signif tissue) before (primary) and after BMI adjustment (BMIadj) are also provided. The asterisk mark attached to the traits indicate which trait remains differentially distributed between at least one of the tissue-specific groups and the remaining population after BMI adjustment. The number of categories for each trait (#categ) are also listed.

| Trait | P adipose | | P brain | | signif tissue | | #categ |
|---|---|---|---|---|---|---|---|
| | primary | BMIadj | primary | BMIadj | primary | BMIadj | |
| *Overall health rating | 4.98E-239 | 1.83E-15 | 6.96E-87 | 1.59E-19 | both | both | 4 |
| *Alcohol intake frequency | 8.05E-133 | 1.51E-06 | 7.47E-73 | 3.09E-11 | both | both | 6 |
| *Frequency of tiredness lethargy in last weeks | 4.04E-91 | 0.63 | 4.03E-38 | 9.44E-06 | both | brain | 4 |
| *Frequency of depressed mood in last 2 weeks | 5.00E-50 | 2.41E-05 | 3.09E-27 | 5.23E-07 | both | both | 4 |
| *Frequency of unenthusiasm disinterest in last 2 weeks | 3.41E-47 | 5.74E-05 | 3.26E-22 | 0.005 | both | adipose | 4 |
| *Falls in the last year | 5.19E-45 | 6.42E-05 | 3.72E-15 | 0.0002 | both | both | 3 |
| Illness injury bereavement stress in last 2 years | 6.49E-44 | 0.5 | 1.11E-10 | 0.93 | both | none | 7 |
| *Alcohol drinker status | 2.50E-35 | 7.56E-18 | 1.66E-29 | 2.17E-24 | both | both | 3 |
| Getting up in morning | 1.05E-28 | 0.14 | 2.20E-23 | 0.003 | both | none | 4 |
| *Weight change compared with 1 year ago | 6.15E-21 | 3.81E-13 | 9.43E-23 | 3.18E-20 | both | both | 3 |
| *Smoking status | 3.14E-06 | 1.31E-08 | 7.62E-24 | 3.06E-11 | both | both | 3 |
| *Sleeplessness insomnia | 7.15E-24 | 5.09E-05 | 2.88E-10 | 0.0001 | both | both | 3 |
| Frequency of tenseness restlessness in last 2 weeks | 5.08E-19 | 0.26 | 5.24E-20 | 0.004 | both | none | 4 |
| Daytime dozing sleeping narcolepsy | 1.51E-17 | 0.84 | 0.0002 | 0.4 | both | none | 4 |
| Blood clot DVT bronchitis emphysema asthma rhinitis eczema allergy diagnosed by doctor | 5.76E-19 | 0.05 | 1.36E-08 | 0.24 | both | none | 6 |
| *Current tobacco smoking | 5.16E-07 | 1.51E-08 | 4.88E-19 | 2.71E-11 | both | both | 3 |
| *Past tobacco smoking | 2.63E-06 | 3.81E-12 | 1.86E-15 | 2.47E-09 | both | both | 4 |
| Qualifications | 1.31E-08 | 0.30 | 1.62E-06 | 0.1 | both | none | 7 |
| Alcohol intake versus 10 years previously | 4.16E-21 | 0.11 | 0.005 | 0.36 | adipose | none | 3 |
| Nap during day | 1.30E-14 | 0.08 | 0.06 | 0.02 | adipose | none | 3 |
| Morning evening person chronotype | 2.88E-07 | 0.43 | 0.0007 | 0.72 | adipose | none | 4 |

For example, *hemoglobin concentration* and *snoring* were heterogeneous for both the adipose and brain groups, *lymphocyte count* and *alcohol intake versus 10 years previously* were heterogeneous only for the adipose group, and *birth weight*, *nervous feelings* only for the brain group (S10 and S12 Tables and Table 2). We observe that *hemoglobin concentration* was lower in individuals from both groups when compared to the population, whereas *reticulocyte percentage* was relatively higher in individuals of the adipose but lower in those with the brain tissue compared to the population (Fig 5 and S10 Table). Among binary traits, *snoring* was more prevalent in those from the adipose group and less prevalent in brain group compared to the population (Fig 6). We observe that for most of the case-control traits, both the tissue-specific groups of individuals had a higher risk of developing the disease compared to the population (Fig 6). Of note, when the tissue-specific relative change of the traits (see Materials and methods) were in the same direction across tissues, they were of different magnitude for a majority of the traits (Figs 5 and 6). For example, the relative change were 15% and 8% for *neutrophil count* (S15 Table). Similar to BMI, we observed phenotypic heterogeneity across individuals with AS (MS) as the prioritized tissue for WHRadjBMI (S5 and S6 Figs and S11, S13 and S14 Tables and S2 Text).

Since BMI itself was differentially distributed between the individuals of the adipose subtype as well as brain subtype compared to the remaining population, we investigated whether the heterogeneity of 84 non-BMI traits (S10 and S12 Tables and Table 2) were induced due to

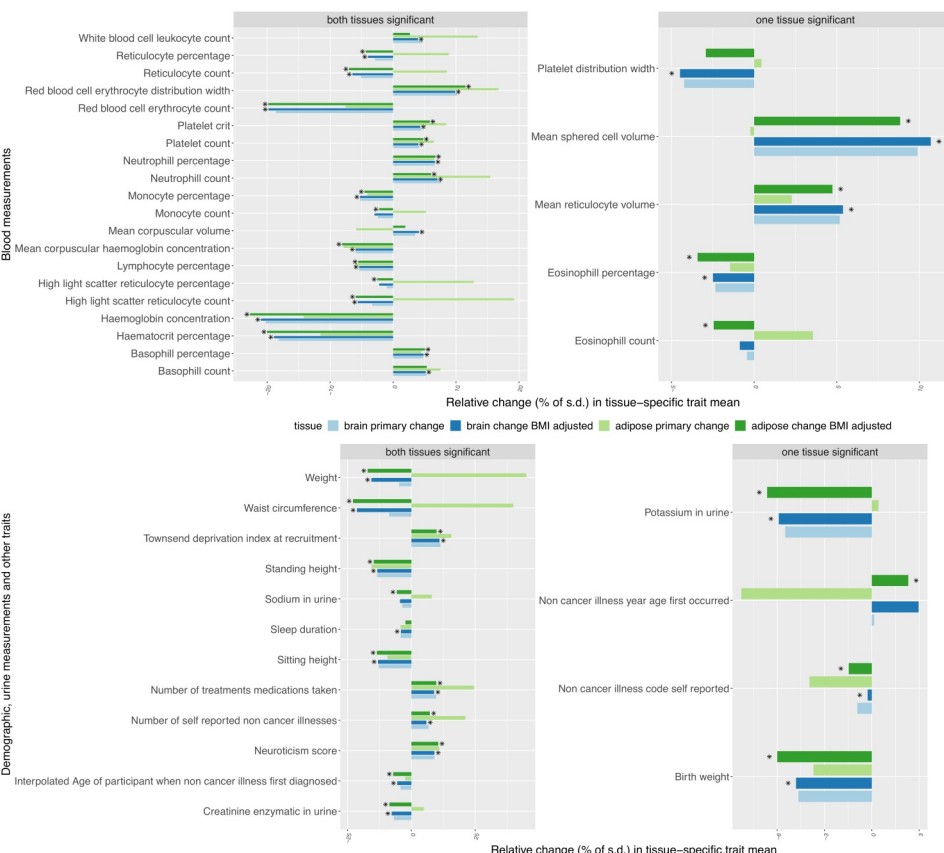

**Fig 5. Percentage of tissue-specific relative change of the quantitative traits that were differentially distributed between the individuals assigned to a tissue-specific subtype of BMI and the remaining population.** Traits in the left panels are primarily heterogeneous for both tissue-specific groups and traits in the right panels are heterogenous for one tissue-specific group. We measure the tissue-specific relative change of a trait by: $\frac{\text{tissue specific mean} - \text{remaining population mean}}{\text{population s.d.}} \times 100$, where the tissue-specific mean is computed only in the individuals with the corresponding tissue-specific subtype. The same measure is calculated for a trait residual obtained after adjusting for BMI to quantify the tissue-specific relative change of the trait after BMI adjustment. The faded green (or blue) bar presents primary adipose (or brain) tissue-specific relative change of a trait compared to the remaining population. The dark green (or blue) bar presents the BMI-adjusted adipose (or brain) specific relative change of a trait. Each trait listed here was found to be differentially distributed between at least one of the adipose or brain specific groups and the remaining population after BMI adjustment. For each trait, the asterisk mark attached to the bars indicates which tissue-specific group remains significantly heterogeneous after BMI adjustment.

BMI heterogeneity (see Materials and methods). After BMI adjustment, 72 (out of 85) traits remained heterogeneous (41 quantitative traits [Fig 5 and S15 Table] and 31 qualitative traits [Fig 6 and Table 2] consistent with unique phenotypic characteristics of these individuals beyond the main phenotype effect. All the quantitative traits which remained heterogeneous after BMI adjustment have the same direction in BMI-adjusted tissue-specific relative change (see Materials and methods) in both adipose and brain compared to the population (Fig 5 and S15 Table). Since we used linear regression while evaluating BMI-adjusted tissue-specific relative change of heterogeneous non-BMI quantitative traits, we also investigated a model-free BMI random matching strategy. We assessed the magnitude of relative change of a trait between individuals with the brain (or adipose) subtype and a group of BMI-matched random individuals drawn from the population (see Materials and methods). For example, the

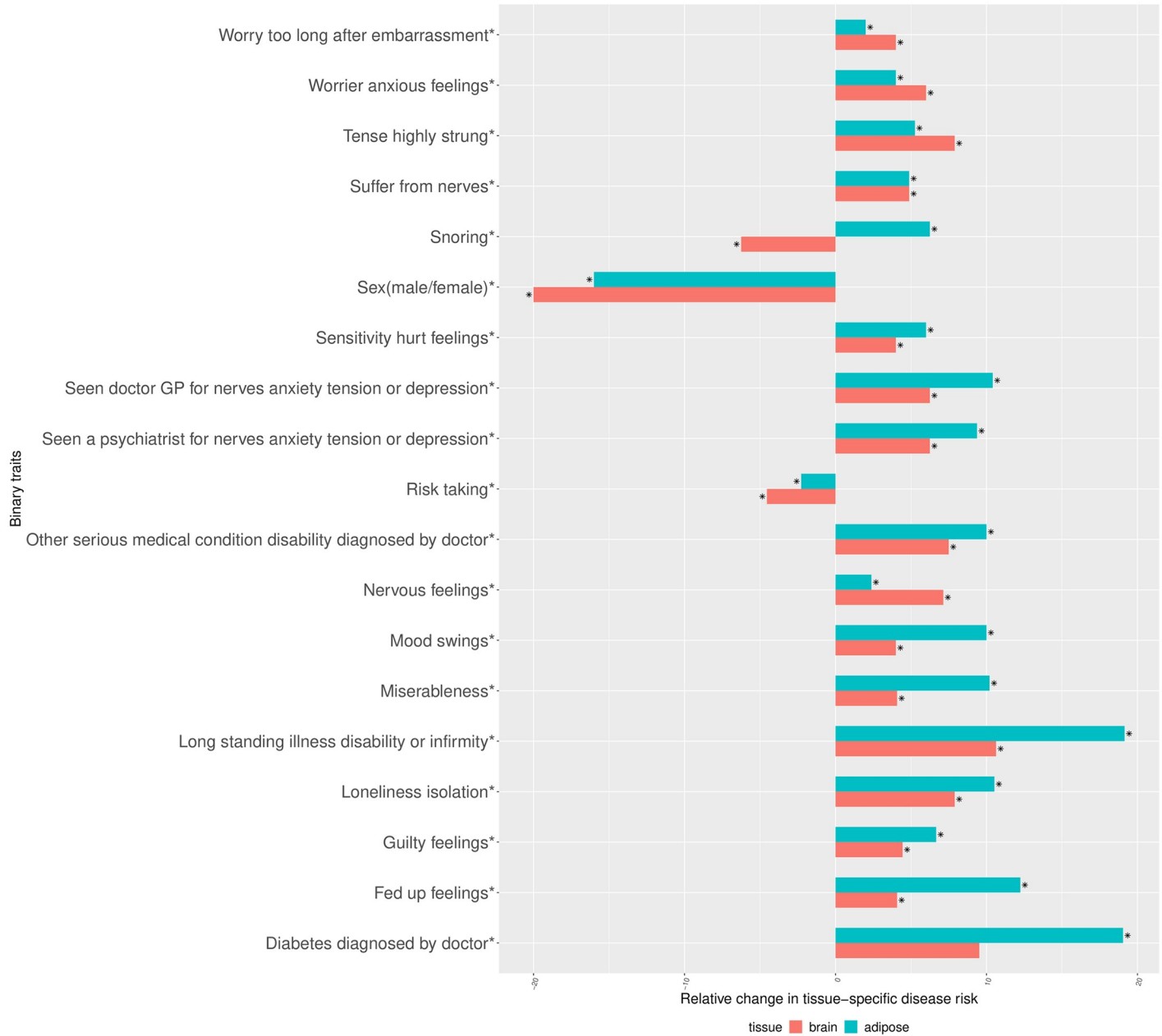

**Fig 6. Percentage of tissue-specific relative change in the risk of case-control traits between the individuals assigned to a tissue-specific subtype of BMI and the population.** The tissue-specific relative change of a disease risk is measured by: $\frac{tissue\ specific\ prevalence\ -\ population\ prevalence}{population\ s.d.} \times 100$. Tissue-specific prevalence of the disorder was computed only in the individuals classified as the corresponding tissue-specific subtype of BMI. The asterisk mark attached to the traits indicate which trait remains differentially distributed between at least one of the adipose and brain tissue-specific groups of individuals and the remaining population after BMI adjustment. For each trait the asterisk mark attached to the bars indicate which tissue-specific group of individuals remains significantly heterogeneous for the trait after BMI adjustment.

magnitude of primary brain-specific relative change (prior to BMI matching) for *hemoglobin concentration* (*mean reticulocyte volume*) decreased from 20% (5%) to 4% (2%) after BMI matching (S17 Table). Of note, it is very difficult to exactly match BMI between a tissue-specific subtype group and the corresponding random group of individuals, because bins of BMI in the tail of its distribution contain very few individuals (S8 Table), the majority of whom

were assigned to a tissue-specific subtype. We observed the same pattern in the results from analogous analyses for WHRadjBMI (S9, S16 and S18 Tables).

To better understand the phenotypic characteristics of the individuals classified to a specific tissue, we performed the following two experiments. First, we shuffled the tissue-specific eQTL SNPs between tissues to create an artificial tissue-specific eQTL set and implemented eGST to identify groups of individuals having subtype specific to the artificial tissues (see Materials and methods). We found that the mean of artificial tissue-specific means of a quantitative trait (found primarily heterogeneous between adipose and/or brain specific group versus the remaining population [S10 Table]) across eQTL shuffles was significantly further from the original corresponding tissue-specific trait mean (S19 Table for BMI and S20 Table for WHRadjBMI). For example, for *waist circumference*, the mean of pseudo tissue-specific means over the random eQTL shuffles is 93.25 for adipose and 91.36 for brain, which are significantly different from the original adipose-specific mean 95.5 ($P < 10^{-100}$) and brain-specific mean 89.2 ($P < 10^{-100}$), respectively (S19 Table). The same pattern was observed for the primary phenotypes BMI (S19 Table) and WHRadjBMI (S20 Table) themselves. Second, we permuted the phenotype data across individuals while keeping the eQTL assignment to tissues fixed as it is in the original data. As before, we also observed that the mean of tissue-specific means of a quantitative trait (found primarily heterogeneous between a tissue-specific group versus the remaining population [S10 Table]) across random phenotype permutations was significantly further from the original corresponding tissue-specific trait mean (S21 Table for BMI and S22 Table for WHRadjBMI).

Finally, we also tested for heterogeneity in distribution of each quantitative trait between the two tissue-specific subtype groups of individuals for BMI, instead of comparing each tissue-specific subtype group with the remaining population. In BMI analysis, we found 27 quantitative traits to be heterogeneously distributed between the adipose-specific group of individuals and the brain-specific group of individuals (S23 Table).

## Computational efficiency

The MAP-EM algorithm underlying eGST is computationally efficient. 70 MAP-EM iterations in the BMI analysis (336$K$ individuals with 1705 adipose-specific eQTLs and 1478 brain-specific eQTLs) took a runtime of 1.75 hours and yielded a log likelihood improvement of $2 \times 10^{-8}$ in the final iteration. Though we ran eGST for a pair of tissues only considering the top eQTL per gene, it is computationally feasible to analyze larger datasets considering more eQTLs and multiple tissues simultaneously.

## Discussion

We proposed a novel approach to quantify tissue-wise genetic contribution to a complex trait and prioritize a relevant tissue for every individual in the study, integrating genotype and phenotype data and an external expression panel data. We applied our method to infer individual-level tissue of interest for BMI and WHRadjBMI in the UK Biobank, integrating expression data in brain, adipose, and muscle tissues from the GTEx consortium, previously shown to be enriched in heritability for these phenotypes [21, 23, 26, 27]. Our approach identified sub-groups of individuals with their genetic susceptibility to the trait mediated in a tissue-specific manner. Interestingly, multiple metabolic traits, neuropsychiatric traits, and other traits attained significant differences between the tissue-specific groups of individuals and the remaining population, suggesting a biologically meaningful interpretation for these sub-groups of individuals. Even after adjusting the traits for the primary phenotype (BMI or

WHRadjBMI), a majority of the traits remained differentially distributed between a tissue-specific group and the remaining population.

We note that the performance of eGST is robust with respect to different reasonable choices of the hyper-parameters in the Bayesian model, mainly due to large sample size considered in contemporary GWAS. For example, we chose the hyper-parameters such that 5% of the total variance of each tissue-specific subtype of the trait is explained due to the corresponding tissue-specific set of SNPs. Since this is a heuristic choice, we experimented with other choices of this prior quantity and observed that a bit of variation in the choice has negligible effect on the final results, which is mainly due to large sample size of the GWAS.

While analyzing the phenotypic heterogeneity, the main reason behind using the 65% threshold of tissue-specific subtype posterior probability was that we obtained a larger number of individuals classified to one of the two tissues compared to using a more stringent 70% threshold. We repeated the analysis for phenotypic characteristics of tissue-specific subtype groups for BMI based on 70% threshold of posterior probability. We observed a similar pattern of phenotypic heterogeneity between a tissue-specific subtype group and the remaining population as compared to using 65% threshold. Using the 70% threshold we found 87 phenotypes in UKB to be heterogeneously distributed compared to 85 phenotypes obtained by using the 65% threshold. The groups of heterogeneous phenotypes are highly overlapping between the two choices. Thus, results on phenotypic heterogeneity were not sensitive to one of these choices. For brevity, we skip providing detailed results on phenotypic heterogeneity obtained by using the 70% threshold. We note that if we consider a much higher choice of the posterior probability threshold, the size of the tissue-specific subtype group will be small. In such a scenario, even though the identified subtype groups will have fewer misclassified individuals, the statistical power to identify heterogeneously distributed phenotypes would decrease mainly due to substantially lower sample size of the subtype groups. For example, using 70%, 80% and 90% thresholds of posterior probability in the BMI analysis, the number of classified individuals was 11871, 2085, and 177, respectively.

We analyzed the UKB sample of individuals, who are unrelated at least up to third degree relatives, i.e., a pair of individuals can be related only as fourth or higher degree relative. To adjust the phenotype for population stratification, we included 20 PCs of genetic ancestries in linear regression. However, fitting a linear mixed model is a more comprehensive strategy to adjust for both population stratification and cryptic relatedness which remain beyond fourth degree relatives in the sample.

The number of classified individuals for the permuted data (phenotype values randomly permuted keeping the genotype data of tissue-specific eQTLs fixed as in the original data) was in the order of thousands. Even though the number of classified individuals was much larger in the original non-permuted data, this result indicates that the false discovery rate of the classified individuals is relatively large. One reason behind this is that we used a less stringent threshold (65%) of tissue-specific subtype posterior probability for the classification. Furthermore, carefully expanding the set of tissue-specific eQTLs may reduce the rate of misclassification.

In real data analysis, we considered the tissues which were reported to be relevant for the phenotype by previous studies. For a different phenotype, if any previous studies have not prioritized the relevant tissues yet, the first step will be to implement the statistical approaches proposed by Finucane et al. [23] and Ongen et al. [22] to detect the tissues important for the phenotype. If at least a pair of tissues are identified to be significantly relevant for the phenotype, we can implement eGST based on the selected tissues. We note that for some complex traits multiple tissues can be biologically relevant. Although in this work we demonstrated the utility of eGST for a pair of tissues for BMI and WHRadjBMI, the MAP-EM algorithm

underlying eGST is general in nature, and can be applied to a larger number ($\geq 2$) of tissues. We note that our model can alternatively be viewed as an approach to assign individuals' phenotypes to a collection of tissues that are biologically important for the trait based on tissue-specific polygenic risk score [38].

We did not explicitly model for individuals who have their genetic contribution to the trait mediated through both tissues. As demonstrated by simulations, such individuals would be assigned posterior probabilities equally distributed across the tissues, and hence would not appear in the tails of the tissue-specific subtype posterior probability distribution. We developed our model under minimal assumptions on tissue-specific genetics. Following previous studies [20, 23] we characterized a tissue by a set of genes specifically over-expressed in it. However, the best possible strategies of choosing an optimal subset of genes (e.g. combination of both over-expressed and low-expressed genes) to efficiently characterize a tissue need to be further investigated. In real data application, we considered the top eQTL of each gene in a tissue mainly for computational convenience. A principled strategy to include more eQTLs of the tissue-specific genes can be to implement COJO in GCTA software [39] to perform conditional and joint analysis of multiple cis-SNPs adjusting for linkage disequilibrium (LD) to identify independent eQTLs for a tissue-specific expressed gene. The set of tissue-specific eQTLs obtained in this way is expected to be larger than the set of top eQTLs only. In future work, we plan to explore how this approach improves the performance of eGST.

We considered top 10% of the tissue-specific expressed genes and the corresponding top eQTLs to implement eGST in UK Biobank. Instead of top 10% genes, we also considered the top 15% of the over-expressed genes in a tissue as the set of tissue-specific expressed genes. In BMI analysis, we ran eGST considering the top eQTLs for the top 15% genes. We observed 73% correlation between the estimate of tissue-specific subtype posterior probability across individuals obtained based on top 10% and 15% tissue-specific expressed genes. If we further increase the percentage of inclusion of tissue-specific genes, it can reduce the tissue-specificity of the selected genes. Thus, how to choose an optimal percentage of tissue-specific genes and corresponding set of eQTLs is a challenging task and requires a separate extensive investigation. However, top 10% expressed genes and the corresponding top eQTLs should always form a core part of the tissue-specific set of genes and eQTLs. Thus, even if we expand the list of tissue-specific expressed genes and corresponding eQTLs, there should be a substantial correlation between the estimates of tissue-specific subtype posterior probability.

In the real data analysis, eGST classified a small percentage of individuals as tissue-specific subtypes. A few possible biological reasons are as follows: 1. a substantial proportion of individuals may have their genetic contribution mediated through both tissues; 2. more than two tissues can be biologically relevant for the phenotype; 3. the set of tissue-specific over expressed genes were considered to represent tissue-specificity, but an interesting possibility is to include lower-expressed genes in a tissue as well.

The estimated tissue-specific subtype heritability across tissues appeared to be small (3% − 4%) for BMI and WHRadjBMI. These estimates should increase upon inclusion of more eQTLs in the analyses. However, we observe that the estimates are comparable to the average estimated heritability of six complex traits (3.4%) due to the effects of imputed gene expressions in blood/adipose (provided in Gusev et al. [16]). Also, a part of the genetic susceptibility for a complex trait is expected to mediate through gene expression.

A recent study [31] has proposed to integrate clinical features related to a disease and imputed gene expression profiles to identify subtypes of the disease. We note that the objective of our study is distinct, and the two approaches are not comparable. Because, we aim to explicitly quantify tissue-wise genetic contribution to the trait at an individual-level and prioritize a relevant tissue for each individual. From a methodological perspective, we proposed an

explicitly likelihood-based classification framework in contrast to their multi-view clustering algorithm.

We conclude with a few caveats and limitations of our work and opportunities for future improvement. We investigated the utility of eGST using adipose and brain tissues for BMI [21, 23, 27, 35–37], and using adipose and muscle tissues for WHRadjBMI [23, 26]. However, the true tissues of interest could be different due to limitations in the existing studies. Since eGST depends on the choice of relevant tissues for a trait, a possible generalization of the model could include an additional mixture component, which does not associate to any of the tissues considered (a null component) and represents individuals for whom none of the tissues is relevant. In BMI analysis, a preliminary experimentation with this model indicates that the subgroup of individuals assigned to the null component remained unclassified (to any of the tissues) by the primary 2-component model. That said, we emphasize that eGST is a general analytic framework that can be applied to a collection of tissues for any complex trait. Even though eGST identified subgroups of individuals having their genetic contribution to the trait mediated in a tissue-specific manner, a major proportion of individuals remained unclassified. Few possible reasons are that we considered two tissues in the analysis, but multiple tissues can be relevant for the trait; we considered the top eQTL for each tissue-specific gene. We note that other types of tissue-specific QTLs (e.g., methylation QTLs, histone QTLs, splicing QTLs, etc. [40]) can also be combined with eQTLs to create a set of SNPs that better represent a tissue-specific genetic architecture.

To explore the performance of eGST in real data analysis, we considered the top eQTLs for the tissue-specific expressed genes. Finucane et al. [23] and Kitsak et al. [41] demonstrated that a promising strategy to comprehend tissue-specificity underlying a complex phenotype is to consider the genes that are specifically expressed in the tissue. Another possibility is to first fine-map the causal eQTLs for each eGene in a tissue [42], and then consider the set of such tissue-specific eQTLs in eGST. In future work, we plan to investigate the merits of this approach.

We developed the model for continuous traits, meaning that to extend the method for case-control data, we would need to use a logistic regression likelihood. Another future methodological investigation is to extend the model under penalized regression framework; if the number of SNPs characterizing the genetic architecture of a tissue becomes large and the ratio between the number of individuals and number of SNPs decreases, model fitting issues can arise. Finally, Fig 1b motivates that if gene expression data across tissues are available, it is possible to use the expression data itself to identify expression subtypes of the trait. However, since expression data is not available in most GWAS cohorts, an alternative avenue will be to impute the genetically regulated component of gene expression, e.g., using PrediXcan [34], EpiXcan [43] and identify tissue of interest based on imputed gene expression. While a possible advantage of such approach will be that all cis eQTLs can be unified to impute tissue-specific expression (instead of top few eQTLs only), a significant noise in the predicted expression due to limited sample size of expression panel data can also trim the improvement in performance. Another limitation of our analysis was that we focused on the GTEx data which does not have a large sample size across tissues. In future work, we plan to apply eGST on other complex traits integrating expression datasets of larger sample size. We provide a user-friendly R software package 'eGST' for general use of our approach: https://cran.r-project.org/web/packages/eGST/index.html.

## Materials and methods

### Model

For simplicity, we describe the model assuming two ($K = 2$) tissues of interest. Suppose, for $n$ unrelated individuals, we have phenotype data $Y = (y_1, \ldots, y_n)$ and expression data for two sets

of tissue-specific expressed genes $E^{(1)}$, $E^{(2)}$ characterizing the two tissues. We define an indicator variable $C$ such that for an individual, $C = k$ iff the genetic susceptibility of the phenotype of the individual is mediated through tissue $k$, $k = 1, 2$ (Fig 1). We model the phenotype of individual $i$ based on the tissue-specific expression of the two sets of tissue-specific genes as:

$$
\begin{aligned}
y_i &= a_1 + \boldsymbol{e}_{1i}^{(1)'}\boldsymbol{b}_1 + \boldsymbol{e}_{2i}^{(1)'}\boldsymbol{d}_1 + \varepsilon_{1i} \text{ if } C_i = 1 \\
&= a_2 + \boldsymbol{e}_{1i}^{(2)'}\boldsymbol{d}_2 + \boldsymbol{e}_{2i}^{(2)'}\boldsymbol{b}_2 + \varepsilon_{2i} \text{ if } C_i = 2
\end{aligned}
\tag{1}
$$

Here, $a_1$ and $a_2$ represent the baseline tissue-specific trait means. $\boldsymbol{e}_{1i}^{(1)}$ and $\boldsymbol{e}_{2i}^{(1)}$ denote the vector of expression values of the first and second tissue-specific set of genes for individual $i$ in the first tissue, respectively; $\boldsymbol{e}_{1i}^{(2)}$ and $\boldsymbol{e}_{2i}^{(2)}$ denote the vector of expression values of the first and second tissue-specific set of genes for individual $i$ in the second tissue. Under $C_i = 1$, $\boldsymbol{b}_1$ and $\boldsymbol{d}_1$ denote the effects of expression of the first and second tissue-specific set of genes in the first tissue on the trait, respectively. Similarly, when $C_i = 2$, $\boldsymbol{b}_2$ and $\boldsymbol{d}_2$ denote the effects of expression of the two gene sets in the second tissue on the trait.

Next we assume that, if $C_i = 1$, the expression of second tissue-specific genes (much low expressed in first tissue) in the first tissue have no effect ($\boldsymbol{d}_1 = \boldsymbol{0}$) on the phenotype of the individual. Similarly, when $C_i = 2$, we assume that the expression of first tissue-specific genes (much low expressed in second tissue) in the second tissue have no effect ($\boldsymbol{d}_2 = \boldsymbol{0}$) on the phenotype. Thus, we obtain the following simplified model under these assumptions:

$$
\begin{aligned}
y_i &\approx a_1 + \boldsymbol{e}_{1i}^{(1)'}\boldsymbol{b}_1 + \varepsilon_{1i} \text{ if } C_i = 1 \\
&\approx a_2 + \boldsymbol{e}_{2i}^{(2)'}\boldsymbol{b}_2 + \varepsilon_{2i} \text{ if } C_i = 2
\end{aligned}
\tag{2}
$$

Expression datasets in general have limited sample size and are not available in large GWAS cohorts. Therefore, we consider genetically regulated component of a tissue-specific gene's expression. However, the genetic component of expression in the GWAS cohort predicted by integrating an external panel of expression data [16, 34] can have substantial noise, which is mainly due to limited sample size of the expression panel (e.g., GTEx). Since eQTLs explain a substantial heritability of gene expression, we use genotypes of tissue-specific eQTLs (i.e., eQTLs for tissue-specific genes) in the GWAS data as a proxy for the predicted genetically regulated component of the expressions of the corresponding tissue-specific genes. Suppose, in a GWAS cohort, we have phenotype data, and genotype data for the two sets of tissue-specific eQTL SNPs corresponding to the two sets of tissue-specific expressed genes, one comprising $m_1$ SNPs and the other comprising $m_2$ SNPs. Then, we consider the following model for the phenotype of individual $i$:

$$
\begin{aligned}
y_i &= \alpha_1 + \boldsymbol{x}_{1i}'\boldsymbol{\beta}_1 + \epsilon_{1i} \text{ if } C_i = 1 \\
&= \alpha_2 + \boldsymbol{x}_{2i}'\boldsymbol{\beta}_2 + \epsilon_{2i} \text{ if } C_i = 2
\end{aligned}
\tag{3}
$$

So, the phenotype of individual $i$ under the tissue of interest $k$ is modeled as $y_i = \alpha_k + \boldsymbol{x}_{ki}'\boldsymbol{\beta}_k + \epsilon_{ki}$, where $\alpha_k$ is the baseline tissue-specific trait mean, $\boldsymbol{x}_{ki}$ is the vector of normalized genotype values of individual $i$ at the eQTL SNPs specific to tissue $k$, $\boldsymbol{\beta}_k = (\beta_{k1}, \beta_{k2} \ldots, \beta_{km_k})$ are their effects on the trait under $C_i = k$, and $\epsilon_{ki}$ is a noise term, $i = 1, \ldots, n$ and $k = 1, 2$. The random errors are distributed as: $\epsilon_{1i} \sim \mathrm{N}(0, \sigma_{\epsilon_1}^2)$ and $\epsilon_{2i} \sim \mathrm{N}(0, \sigma_{\epsilon_2}^2)$. The above mixture model can be viewed as a variant of finite mixture of regression models where each component is a linear model with a distinct set of predictors.

Of note, the mixture model in our context is identifiable because the mean parameter in each component is a function of the genotype vector of the set of tissue-specific eQTLs, which is distinct across tissues [44].

**Prior distributions.** $P(C_i = k) = w_k$ is the prior proportion of individuals for whom the phenotype has $k^{th}$ tissue-specific genetic effect. We assume that the eQTL SNP sets across $k$ tissues are non-overlapping and that each element in $\boldsymbol{\beta}_k$, the genetic effect of $k^{th}$ tissue-specific eQTLs on the trait, is independently drawn from $N(0, \sigma^2_{x_k})$. If $\sigma^2_{y_k}$ is the variance of the trait under $C_i = k$, then $\sigma^2_{x_k} = \frac{h^2_k \sigma^2_{y_k}}{m_k}$, where $h^2_k$ is the heritability of the trait under $C_i = k$ due to $k^{th}$ tissue-specific $m_k$ eQTLs, and is termed as $k^{th}$ tissue-specific subtype heritability. We also assume that $\alpha_1 \sim N(0, \sigma^2_\alpha)$ and $\alpha_2 \sim N(0, \sigma^2_\alpha)$, with fixed $\sigma^2_\alpha = 1$. For $K = 2$, we assume that $w_1 \sim$ Beta $(s_1, s_2)$ [$w_2 = 1 - w_1$], which will be a Dirichlet distribution for more than two tissues. We consider fixed values of $s_1 = s_2 = 1$. Next, we assume: $\sigma^2_{x_1}$ and $\sigma^2_{x_2} \sim$ Inverse−Gamma$(a_x, b_x)$; $\sigma^2_{\epsilon_1}$ and $\sigma^2_{\epsilon_2} \sim$ Inverse−Gamma$(a_\epsilon, b_\epsilon)$. We choose fixed values of $a_x, b_x, a_\epsilon, b_\epsilon$ such that in the prior expectation, 5% of the total variance of each tissue-specific subtype (under $C_i = 1$ or 2) of the trait is explained by the corresponding set of tissue-specific eQTL SNPs and 95% of the variance remains unexplained.

## Inference procedure

Under this Bayesian framework, we implemented the maximum a posteriori (MAP) expectation-maximization (EM) algorithm (Algorithm 1) to estimate the posterior probability that the phenotype of individual $i$ is mediated through the genetic effects of eQTLs specific to tissue $k$ ($P(C_i = k|X, Y)$). We note that it is also possible to consider a frequentist framework of the mixture model, i.e., instead of having a distribution, $(w_1, w_2)$, $(\alpha_1, \alpha_2)$, $(\boldsymbol{\beta}_1, \boldsymbol{\beta}_2)$, $(\sigma^2_{\epsilon_1}, \sigma^2_{\epsilon_2})$ can be assumed to have a fixed unknown true value. We implemented an EM algorithm to estimate the tissue-specific posterior probability across individuals under the frequentist framework. Next, for a general $K (\geq 2)$ number of tissues, we outline the MAP-EM algorithm that implements the Bayesian framework, and the EM algorithm that implements the frequentist framework of the mixture model [45, 46].

For individual $i$ and tissue $k$, $i = 1, \ldots, n$ and $k = 1, \ldots, K$, $P(C_i = k) = w_k$; $\Sigma_k w_k = 1$. Denote $k^{th}$ tissue-specific set of parameters by $\theta_k = (w_k, \alpha_k, \boldsymbol{\beta}_k, \sigma^2_{x_k}, \sigma^2_{\epsilon_k})$ and full set of parameters by $\Theta = (\theta_1, \ldots, \theta_K)$. Under the mixture model, the likelihood of individual $i$ takes the following form:

$$f(y_i|\Theta) = \sum_{k=1}^{K} w_k \phi(y_i|\mu = \mu_{ik}, \sigma^2 = \sigma^2_{\epsilon_k}); \quad \text{with} \quad \mu_{ik} = \alpha_k + \boldsymbol{x}'_{ki}\boldsymbol{\beta}_k, \tag{4}$$

where $\phi(.|.)$ denotes the normal density. Thus, the full data log-likelihood conditioned on $\Theta$ is given by: $\log f(Y|\Theta) = \sum_{i=1}^{n} \log f(y_i|\Theta)$. The prior log-likelihood of $(C_1, \ldots, C_n)$ is given by $\sum_{i=1}^{n} \log f(C_i)$, where $f(C_i)$ is: $P(C_i = k) = w_k$; $(w_1, \ldots, w_K) \sim$ Dirichlet$(s_1, \ldots, s_K)$. The prior of $k^{th}$ tissue-specific parameters $\theta_k$ has the following hierarchical structure: $\boldsymbol{\beta}_k|\sigma^2_{x_k} \sim N_{m_k}(0, \sigma^2_{x_k} I_{m_k})$, $\sigma^2_{x_k} \sim$ Inverse − Gamma$(a_k, \boldsymbol{b}_k)$, $\sigma^2_{\epsilon_k} \sim$ Inverse − Gamma$(a_\epsilon, \boldsymbol{b}_\epsilon)$, $\alpha_k \sim N(0, \sigma^2_\alpha)$. In the prior, $\theta_1, \ldots, \theta_K$ are independently distributed. Define the posterior probability that the phenotype of individual $i$ be assigned tissue $k$ as:

$$\gamma_{ik} = P(C_i = k|y_i, \Theta) = \frac{w_k \phi(y_i|\mu_{ik}, \sigma^2_{\epsilon_k})}{\sum_{t=1}^{K} w_t \phi(y_i|\mu_{it}, \sigma^2_{\epsilon_t})}; \quad \sum_{k=1}^{K} \gamma_{ik} = 1 \tag{5}$$

Thus, the choice of the tissue of interest across individuals is quantified by $\Gamma = \{\gamma_{ik}; i = 1, \ldots, n; k = 1, \ldots, K\}$. Next, we define the total membership weight of $k^{th}$ tissue-specific subtype: $n_k = \sum_{i=1}^{n} \gamma_{ik}, \Sigma_k n_k = n$. In the expectation-maximization algorithm, the main component which we maximize is given by: $Q(\Theta|\Theta^{(r)}) = \sum_{i=1}^{n} Q_i(\Theta|\Theta^{(r)})$, where the conditional expectation $Q_i(\Theta|\Theta^{(r)}) = E_{C_i}|y_i, \Theta^{(r)}\{\log f(y_i, C_i|\Theta)\} = \sum_{k=1}^{K} \gamma_{ik}[\log w_k + \log \phi(y_i|\mu_{ik}, \sigma^2_{\epsilon_k})]$. To obtain $\Theta^{(r+1)}$, we maximize $\{Q(\Theta|\Theta^{(r)}) + \log f(\Theta)\}$ in the MAP-EM algorithm implementing the Bayesian framework, and maximize only $Q(\Theta|\Theta^{(r)})$ in the EM algorithm implementing the frequentist framework [45, 46]. The steps of the MAP-EM and EM algorithm are provided in Algorithm 1 and 2, respectively. Our main inference is based on the posterior probability matrix $\Gamma = \{\gamma_{ik}; i = 1, \ldots, n; k = 1, \ldots, K\}$. For example, $\gamma_{i1} > 65\%$ indicates that tissue 1 is likely to be the tissue of interest for individual $i$. We also designed a MCMC algorithm to implement the eGST model which we describe in S1 Algorithm in S1 Text.

**Algorithm 1** Maximum a posteriori (MAP) expectation maximization (EM) algorithm under Bayesian framework of the mixture model

```
1: Initialization: For k = 1, ..., K, choose σ²⁽⁰⁾_εₖ = 0.95 × var(Y),
```
$\sigma^{2(0)}_{x_k} = \frac{0.05 \times var(Y)}{m_k}$, $\alpha^{(0)}_k = 0$, $w^{(0)}_k = \frac{1}{K}$; and simulate $\boldsymbol{\beta}^{(0)}_k$ from $N(0, \sigma^{2(0)}_{x_k} I_{m_k})$. Compute the initial log-likelihood: $\log L^{(0)} = \frac{1}{n}\sum_{i=1}^{n} \log\left(\sum_{k=1}^{K} w^{(0)}_k \phi(y_i|\mu = \mu^{(0)}_{ik}, \sigma^2 = \sigma^{2(0)}_{\epsilon_k})\right)$. Next, for iteration $r = 0, 1, \ldots$
```
2: E-step: Compute:
```

$$\gamma^{(r)}_{ik} = \frac{w^{(r)}_k \phi(y_i|\mu^{(r)}_{ik}, \sigma^{2(r)}_{\epsilon_k})}{\sum_{t=1}^{K} w^{(r)}_t \phi(y_i|\mu^{(r)}_{it}, \sigma^{2(r)}_{\epsilon_t})}; \quad \mu_{ik} = \alpha_k + \boldsymbol{x}'_{ki}\boldsymbol{\beta}_k; \quad i = 1, \ldots, n; \quad k = 1, \ldots, K$$

$$n^{(r)}_k = \sum_{i=1}^{n} \gamma^{(r)}_{ik}, \quad k = 1, \ldots, K$$

```
3: M-step: For k = 1, ..., K, update:
```

$$w^{(r+1)}_k = \frac{n^{(r)}_k}{n}$$

$$\alpha^{(r+1)}_k = \frac{1}{n^{(r)}_k + \frac{\sigma^{2(r)}_{\epsilon_k}}{\sigma^2_{\alpha}}} \times \sum_{i=1}^{n} \gamma^{(r)}_{ik}(y_i - \boldsymbol{x}'_{ki}\boldsymbol{\beta}^{(r)}_k)$$

$$\boldsymbol{\beta}^{(r+1)}_k = \left(\sum_{i=1}^{n} \gamma^{(r)}_{ik}\boldsymbol{x}_{ki}\boldsymbol{x}'_{ki} + \frac{\sigma^{2(r)}_{\epsilon_k}}{\sigma^{2(r)}_{x_k}} I_{m_k}\right)^{-1} \sum_{i=1}^{n} \gamma^{(r)}_{ik}(y_i - \alpha^{(r)}_k)\boldsymbol{x}_{ki}$$

$$\sigma^{2(r+1)}_{\epsilon_k} = \frac{1}{n^{(r)}_k + 2a_{\epsilon} + 1}\left[2b_{\epsilon} + \sum_{i=1}^{n} \gamma^{(r)}_{ik}(y_i - \mu^{(r)}_{ik})^2\right]$$

$$\sigma^{2(r+1)}_{x_k} = \frac{\boldsymbol{\beta}'^{(r)}_k\boldsymbol{\beta}^{(r)}_k + 2b_x}{m_k + 2a_x + 1}$$

4: **Convergence check:** Compute the new log-likelihood:

$$\mathrm{logL}^{(r+1)} = \frac{1}{n}\sum_{i=1}^{n}\log\left(\sum_{k=1}^{K}w_k^{(r+1)}\phi(y_i|\mu_{ik}^{(r+1)},\sigma_{\epsilon_k}^{2^{(r+1)}})\right)$$

**Return to step 2**, if $|\mathrm{logL}^{(r+1)} - \mathrm{logL}^{(r)}| > \delta$, for a pre-fixed threshold $\delta$ (e.g. $10^{-5}$).

**Algorithm 2** Expectation Maximization (EM) algorithm under frequentist framework of the mixture model

1: **Initialization:** For $k = 1, \ldots, K$, choose $\sigma_{\epsilon_k}^{2^{(0)}} = 0.95 \times var(Y)$, $\sigma_{x_k}^{2^{(0)}} = \frac{0.05 \times var(Y)}{m_k}$, $\alpha_k^{(0)} = 0$, $w_k^{(0)} = \frac{1}{K}$; and simulate $\boldsymbol{\beta}_k^{(0)}$ from $\mathrm{N}(0, \sigma_{x_k}^{2^{(0)}}I_{m_k})$. Compute the initial log-likelihood: $\mathrm{logL}^{(0)} = \frac{1}{n}\sum_{i=1}^{n}\log\left(\sum_{k=1}^{K}w_k^{(0)}\phi(y_i|\mu = \mu_{ik}^{(0)}, \sigma^2 = \sigma_{\epsilon_k}^{2^{(0)}})\right)$. Next, for iteration $r = 0, 1, \ldots$

2: **E-step:** Compute:

$$\gamma_{ik}^{(r)} = \frac{w_k^{(r)}\phi(y_i|\mu_{ik}^{(r)},\sigma_{\epsilon_k}^{2^{(r)}})}{\sum_{t=1}^{K}w_t^{(r)}\phi(y_i|\mu_{it}^{(r)},\sigma_{\epsilon_t}^{2^{(r)}})}; \quad \mu_{ik} = \alpha_k + \boldsymbol{x}_{ki}'\boldsymbol{\beta}_k; \quad i = 1, \ldots, n; \quad k = 1, \ldots, K$$

$$n_k^{(r)} = \sum_{i=1}^{n}\gamma_{ik}^{(r)}, \quad k = 1, \ldots, K$$

3: **M-step:** For $k = 1, \ldots, K$, update:

$$w_k^{(r+1)} = \frac{n_k^{(r)}}{n}$$

$$\alpha_k^{(r+1)} = \frac{1}{n_k^{(r)}} \times \sum_{i=1}^{n}\gamma_{ik}^{(r)}(y_i - \boldsymbol{x}_{ki}'\boldsymbol{\beta}_k^{(r)})$$

$$\boldsymbol{\beta}_k^{(r+1)} = \left(\sum_{i=1}^{n}\gamma_{ik}^{(r)}\boldsymbol{x}_{ki}\boldsymbol{x}_{ki}'\right)^{-1}\sum_{i=1}^{n}\gamma_{ik}^{(r)}(y_i - \alpha_k^{(r)})x_{ki}$$

$$\sigma_{\epsilon_k}^{2^{(r+1)}} = \frac{1}{n_k^{(r)}}\sum_{i=1}^{n}\gamma_{ik}^{(r)}(y_i - \mu_{ik}^{(r)})^2$$

4: **Convergence check:** Compute the new log-likelihood:

$$\mathrm{logL}^{(r+1)} = \frac{1}{n}\sum_{i=1}^{n}\log\left(\sum_{k=1}^{K}w_k^{(r+1)}\phi(y_i|\mu_{ik}^{(r+1)},\sigma_{\epsilon_k}^{2^{(r+1)}})\right)$$

**Return to step 2**, if $|\mathrm{logL}^{(r+1)} - \mathrm{logL}^{(r)}| > \delta$, for a pre-fixed threshold $\delta$.

## Simulation design and choice of parameters

Consider $n$ individuals and two non-overlapping sets of $m_1$ SNPs and $m_2$ SNPs representing eQTL SNP sets specific to two tissues. We chose the SNPs on chromosome $8 - 17$ from the array SNPs in the UK Biobank (UKB). We pruned for LD between the SNPs such that two

consecutive SNPs (on a chromosome) included in a SNP set had $r^2 < 0.25$ (based on UKB in-sample LD). Each SNP had MAF >1% and satisfied Hardy Weinberg Equilibrium (HWE). We collected genotype data at both sets of SNPs for $n$ individuals that were randomly selected from 337,205 white-British individuals in the UKB.

Let $w = (w_1, w_2)$ denote the proportions of individuals in the sample assigned to the two tissues where $(100 \times w_1)$% individuals are assigned the first tissue-specific subtype and $(100 \times w_2)$% individuals are assigned the second tissue-specific subtype. We assume that $m_k$ SNPs explain $(100 \times h_k^2)$% of the total variance of $k^{th}$ tissue-specific subtype, $k = 1, 2$. So $h_k^2$ is the heritability of $k^{th}$ tissue-specific subtype of the trait due to $m_k$ SNPs representing $k^{th}$ tissue-specific eQTLs, $k = 1, 2$. Thus, if first subtype of $Y$ has a total variance $\sigma_{y_1}^2$, we draw each element of $\boldsymbol{\beta}_1$ as: $\beta_{1j} \sim \mathrm{N}\left(0, \frac{h_1^2 \sigma_{y_1}^2}{m_1}\right)$, $j = 1, \ldots, m_1$. Similarly we simulate $\boldsymbol{\beta}_2$, the genetic effect of second set of $m_2$ SNPs on the second subtype of $Y$. For simplicity, we assume $\sigma_{y_1}^2 = \sigma_{y_2}^2 = \sigma_y^2$, but the performance of eGST remains similar for other choices of this parameter. If the genetic susceptibility of an individual's phenotype was assigned to be mediated through first tissue, we simulated the phenotype as: $y = \alpha_1 + x_1' \boldsymbol{\beta}_1 + \epsilon_1$, where $x_1$ is the normalized genotype values of the individual at the first set of SNPs. While simulating the phenotype, we normalized the genotypes at each of first tissue-specific $m_1$ SNPs only based on the individuals assigned to the first tissue-specific subtype. However, when applying eGST on a simulated dataset, we normalized the genotypes at each SNP based on all $n$ individuals in the sample, because the tissue of interest across individuals are unknown. The random error components have the following distribution: $\epsilon_1 \sim \mathrm{N}(0, (1 - h_1^2)\sigma_{y_1}^2)$ and $\epsilon_2 \sim \mathrm{N}(0, (1 - h_2^2)\sigma_{y_2}^2)$.

We varied the choice of parameters to evaluate eGST in various simulation scenarios. We chose $\sigma_y^2 = 10$, and initially assumed $\alpha_1 = \alpha_2 = 0$ and simulate $\boldsymbol{\beta}_1, \boldsymbol{\beta}_2$ from zero-mean normal distributions. We considered all possible combinations of $(w_1, w_2)$ where $w_1 \in \left(\frac{1}{2}, \frac{1}{3}\right)$ and $w_2 \in \left(\frac{1}{2}, \frac{1}{3}\right)$, and all possible combinations of $(h_1^2, h_2^2)$, where $h_1^2$ and $h_2^2 \in (10\%, 20\%, 30\%, 40\%, 50\%)$. We also considered two unrealistic scenarios of null and high subtype heritability: $h_1^2 = h_2^2 = 0\%$ and $h_1^2 = h_2^2 = 90\%$ to evaluate if eGST is performing as expected in these extreme scenarios. We chose $(m_1, m_2)$ with $m_1 = 1000, 1500$ and $m_2 = 1000, 1500$. Initially we chose $n = 40,000$, and later $n = 100,000$ to explore the effects of an increased sample size. For each choice of the complete set of simulation parameters, we summarized the results of eGST across 50 simulated datasets. We also performed simulations for $\alpha_1 \neq \alpha_2$ and different non-zero mean of $\boldsymbol{\beta}_1, \boldsymbol{\beta}_2$ distributions.

## BMI and WHRadjBMI analysis in the UK Biobank integrating GTEx data

We implemented eGST to infer the individual-level tissue of interest for two obesity related measures, BMI and WHRadjBMI, in the UK Biobank [32, 33], integrating expression data from the Genotype-Tissue Expression (GTEx) project [17, 18]. Sets of tissue-specific expressed genes were obtained from Finucane et al. [23] who analyzed the GTEx data and considered a gene to be specifically expressed in a tissue of interest if the gene's mean expression in the tissue is substantially higher than its mean expression in other tissues combined, and calculated a t-statistic to rank the genes with respect to higher expression in a specific tissue. Similar to their work [23], we considered the top 10% of all genes (2485 such genes) in a tissue, ranked according to descending value of the tissue-specific t-statistic, as the set of genes specifically expressed in the tissue.

We focused on the adipose and brain tissue for BMI, and the adipose and muscle tissue for WHRadjBMI. We took the union of the sets of genes specifically expressed in adipose

subcutaneous and adipose visceral tissues, and considered it as the adipose-specific gene set. Similarly, we took the union of sets of genes specifically expressed in the brain cerebellum and brain cortex regions (these two had maximum sample size among different brain regions) to create a brain-specific set of genes. We excluded the genes overlapping between these two sets to consider non-overlapping sets of adipose and brain specific genes. For WHRadjBMI, we considered adipose subcutaneous and muscle skeletal connective tissue, and excluded the genes overlapping between the two sets of top 10% expressed genes within the tissues. We considered genes on the autosomal chromosomes 1–22. In BMI analysis, the main reason behind merging two type of adipose (or brain) tissues together to represent adipose (or brain) was to increase the number of tissue-specific eGenes per tissue. For WHRadjBMI analysis, we considered the adipose subcutaneous and muscle skeletal tissues to find different possible patterns in the performance of eGST that might be missed in BMI analysis due to merging tissues.

The subsets of primary sets of tissue-specific genes that were found to be eGenes in GTEx were included in subsequent analyses. A gene is considered to be an eGene if at least one cis-SNP is significantly associated with its expression at FDR level 0.05 [17]. For WHRadjBMI analysis, among the initially selected 2228 adipose subcutaneous tissue-specific genes, 1152 genes were found to be eGenes for which at least one bi-allelic SNP was reported to be an eQTL in the GTEx summary-level data (version 7). Similarly, we had 1272 eGenes for muscle skeletal tissue. In BMI analysis, we had 1887 eGenes for adipose and 1653 eGenes for brain. For each gene in a tissue, we took the top bi-allelic eQTL SNP (smallest SNP-expression association p-value) with MAF >1%. In BMI analysis, while creating an adipose-specific set of eQTLs, if a gene was both adipose subcutaneous and visceral tissue-specific gene, we included the top eQTL of the gene in both tissues, one in subcutaneous and one in visceral. We implemented the same strategy for brain tissue, as well.

Next, we obtained the subset of SNPs from each set of tissue-specific eQTL SNPs (obtained from GTEx), which were genotyped or imputed in UKB (imputation accuracy score > 0.9). The SNPs were also screened for HWE (p-value $> 10^{-6}$) in UKB. We LD-pruned each set of tissue-specific eQTL SNPs based on $r^2$ threshold 0.25 using UKB in-sample LD. In a tissue-specific set, if two eQTL SNPs had $r^2 > 0.25$, we excluded the one for which the minimum of SNP-expression association p-value (in GTEx) across the genes (for which it was found to be the top eQTL) was larger. Finally, after LD pruning, we had 1705 eQTL SNPs specific to adipose and 1478 eQTL SNPs specific to brain for BMI analysis. We obtained 953 eQTL SNPs specific to adipose subcutaneous tissue and 1052 eQTL SNPs specific to muscle skeletal tissue for WHRadjBMI analysis. We used individual-level genotype data for the tissue-specific SNP sets in UKB to infer tissue of interest across individuals. Before running eGST, we normalized genotypes at each SNP in the two tissue-specific sets based on the whole sample of individuals.

**Phenotype data.**   We considered the BMI of 337,205 unrelated white-British individuals in the UK Biobank (full release) and excluded individuals for whom BMI or relevant covariates (age, sex, etc.) were missing. We then adjusted BMI for age, sex, and the top 20 principal components (PCs) of genetic ancestry by linear regression and obtained the BMI residuals. We initially developed eGST assuming that each tissue-specific subtype of the trait follows a normal distribution. Since the BMI residuals obtained after the adjustment of covariates deviated substantially from the normal distribution (p-value of Kolmogorov Smirnov (KS) test for deviation from normal distribution $< 2.2 \times 10^{-16}$), we applied the rank-based inverse normal transformation on the BMI residuals, and implemented eGST for the transformed phenotype data. We adjusted WHR for BMI to obtain WHRadjBMI. We then adjusted WHRadjBMI for age, sex, and top 20 PCs of genetic ancestry. Since the WHRadjBMI residuals significantly deviated from the normal distribution, we applied the inverse normal transformation on the residuals.

**Genetic characteristics.** We contrasted the genetic basis of the groups of individuals assigned to the adipose and brain-specific subtypes of BMI. Let $\boldsymbol{\beta}_1$ denote the joint SNP-effects of the adipose-specific eQTLs on the BMI of the individuals classified as the adipose-specific subtype for whom the adipose-specific posterior probability obtained by eGST was > 50%. We chose a relaxed threshold of posterior probability because we used multiple linear regression (MLR) to estimate the joint SNP effects of a set of tissue-specific eQTLs on the BMI of a tissue-specific group of individuals, and MLR requires sufficiently large number of individuals (assigned to the corresponding tissue-specific subtype) in the sample for efficient estimation of the model parameters. Thus, if $Y_1 = \{Y_i: C_i = 1\}$ and $X_1$ is the genotype matrix of adipose eQTLs for these adipose-specific individuals, we fit the linear model: $E(Y_1) = X_1\boldsymbol{\beta}_1$. Let $\gamma_1$ be the joint SNP effects of the brain eQTLs on BMI of individuals assigned to the adipose subtype. Since the BMI of individuals assigned to the adipose subtype should have larger effects from adipose-specific eQTLs than from brain-specific eQTLs, we should expect that the magnitude of a general element in $\boldsymbol{\beta}_1$ would be larger than the magnitude of a general element in $\gamma_1$. Based on the individuals assigned to the adipose subtype, we estimated $\boldsymbol{\beta}_1$ and $\gamma_1$ using multiple linear regression of BMI residual on the genotypes of adipose eQTLs and brain eQTLs in UKB, respectively. Based on the estimated $\boldsymbol{\beta}_1$ and $\gamma_1$ vectors, we performed the non-parametric Wilcoxon rank sum (WRS) test to evaluate $H_0: |\beta_1| = |\gamma_1|$ versus $H_1: |\beta_1| > |\gamma_1|$, where $\beta_1$ and $\gamma_1$ represent a general element in $\boldsymbol{\beta}_1$ and $\gamma_1$ vectors, respectively. Similarly, we tested if the magnitude of the effect of a brain eQTL SNP on the BMI of individuals assigned to brain-specific subtype $(\beta_2)$ was larger than the corresponding effect magnitude of an adipose eQTL SNP $(\gamma_2)$. We also tested whether the adipose eQTLs had a larger SNP effect on the adipose subtype of BMI than on the brain subtype, and whether brain eQTLs had a larger effect on the brain subtype than on the adipose subtype. We performed the analogous experiments for the groups of individuals assigned to AS and MS tissue-specific subtype of WHRadjBMI.

**Phenotypic characteristics.** We explored if the group of individuals whose BMI were classified as a tissue-specific genetic subtype is phenotypically distinct from the rest of the population, with respect to various other phenotypes collected in the UK Biobank. We considered 106 such phenotypes and individually tested each trait for being differentially distributed between individuals of each tissue-specific subtype and the remaining population (for BMI, 11,838 individuals assigned to adipose subtype and 13,354 individuals assigned to brain subtype based on 65% threshold of subtype posterior probability [S1 Table]). We performed the Wilcoxon rank sum (WRS) test for a quantitative trait and $\chi^2$ test based on the contingency table for a qualitative/categorical trait. We corrected the p-values for multiple testing across traits using the Bonferroni correction procedure. The same approach was adopted to find the traits differentially distributed between individuals classified as a tissue-specific subtype of WHRadjBMI and the remaining population (for WHRadjBMI, 11,803 individuals with AS subtype and 7,238 individuals of MS subtype [S1 Table]). For a binary/case-control trait, we term the percentage of individuals (among those assigned to the tissue) who had the disorder as tissue-specific risk of the disease.

**A random group of individuals is phenotypically homogeneous with the remaining population.** For BMI, we randomly selected two groups of individuals from the population with the same size as the groups of tissue-specific BMI subtype (11,838 and 13,354) and evaluated phenotypic heterogeneity across 106 traits between each of the two random groups and the rest of the population using WRS test for a continuous trait and contingency table $\chi^2$ test for a qualitative trait (as before). We repeated the random selection of individuals from the population to replicate the experiment. We did the same experiment for WHRadjBMI.

**BMI (or WHRadjBMI) adjusted phenotypic heterogeneity.** In the above analysis for BMI, BMI itself was found to be differentially distributed between the individuals with the

adipose (as well as brain) specific subtype and the remaining population. Therefore, we further investigated whether the heterogeneity of non-BMI traits between a subtype group and the remaining population were induced due to BMI heterogeneity. For each quantitative trait initially found to be heterogeneous between individuals assigned to one of the subtype groups and the remaining population (S10 Table), we first adjusted the trait for BMI in the whole population and obtained the trait residuals. We then tested for heterogeneity between the trait residual in the adipose (or the brain) subtype group and the remaining population using WRS test. Similarly for qualitative/categorical traits that were initially heterogeneous (Fig 6 and Table 2 and S12 Table), we performed a binomial or multinomial (depending on the number of categories of the trait) logistic regression adjusting for BMI in the population. We adopted the same strategy for WHRadjBMI (which itself was found to be differentially distributed between AS as well as MS group and the remaining population) to find which among the non-WHR traits remain heterogeneous after WHRadjBMI adjustment.

**Tissue-specific relative change.** For each quantitative trait that was differentially distributed between the individuals of a tissue-specific subtype and the remaining population, we measured the relative change (or difference) of the trait between the tissue-specific subtype group and the remaining population as: $\frac{tissue\_specific\_mean\ -\ remaining\_population\_mean}{population\ s.d.} \times 100$, where the tissue-specific mean of the trait is calculated only in the individuals classified as the corresponding tissue-specific subtype of BMI (or WHRadjBMI). To quantify BMI-adjusted tissue-specific relative change of a primarily heterogeneous quantitative trait, we computed the same measure for BMI-adjusted trait residual (instead of the trait itself). To evaluate the tissue-specific relative change in the risk of a binary/case-control trait, we calculated $\frac{tissue\_specific\_prevalence\ -\ population\_prevalence}{population\ s.d.} \times 100$, where the tissue-specific prevalence of a disease is computed only in the individuals assigned to the corresponding tissue-specific subtype.

**BMI (or WHRadjBMI) matched tissue-specific relative change.** In order to further investigate the role of tissue-specific genetics (uncoupled from the role of BMI heterogeneity) underlying the phenotypic characteristics of the individuals assigned to a tissue-specific subtype of BMI, we performed the following experiment. We split the range of BMI of the individuals assigned to the adipose subtype (11,838 individuals [S1 Table]) into 30 consecutive non-overlapping bins. In each BMI bin, we counted the number of individuals assigned to the adipose subtype, and randomly sampled the same number of individuals from all of the individuals contained in the bin. In this way, we randomly selected a pool of individuals (with the same size as the adipose-specific group) from the population, who are matched with the BMI of the adipose subtype individuals. Next, for each non-BMI quantitative trait which was found to be heterogeneous between the adipose group and the remaining population after BMI adjustment (S15 Table), we computed: $\left| \frac{adipose\_specific\_mean - BMI\_matched\_random\_mean}{population\ s.d.} \right| \times 100$, where the adipose specific mean of the trait is calculated only in the individuals of the adipose subtype and the BMI matched random mean is the trait mean calculated only in the BMI matched (with adipose group) random pool of individuals. This measure quantifies the relative change/difference of the trait between the individuals assigned to the adipose subtype and the corresponding BMI-matched random individuals selected from the population. This should provide insights into the phenotypic characteristics of the individuals with the adipose subtype, which is solely mediated through adipose-specific genetics (uncoupled from the corresponding effect of BMI heterogeneity between the adipose group and the remaining population). We repeated the random selection of BMI-matched individuals 500 times and computed the mean and s.d. of the above measure of BMI-matched tissue-specific relative change of a quantitative trait across random selections. We replicated the same experiment for individuals with brain subtype of BMI. For WHRadjBMI, we performed the same experiment to characterize the phenotypic

characteristics of AS (or MS) subtype group induced due to AS- (or MS-) specific genetics only.

**Tissue-specificity of phenotypic characteristics.** To investigate tissue-specificity of the phenotypic characteristics of the individuals assigned to adipose and brain specific subtype of BMI, we randomly shuffled/exchanged 739 (half of the minimum of number of adipose and brain specific eQTLs $= \frac{1478}{2}$) eQTLs between the set of adipose and brain specific eQTLs to create artificial tissue-specific eQTL sets. We considered 500 such random shuffles. Keeping the phenotype data fixed, for the genotype data at each set of artificial tissue-specific eQTLs, we ran eGST to identify the groups of individuals with the BMI subtype specific to the artificial adipose and brain tissues (based on the posterior probability threshold of 65%). Next, for each quantitative trait that was found to be primarily heterogeneous between the individuals assigned to the original adipose (or brain) subtype of BMI and the remaining population (S10 Table), we computed the artificial adipose and brain tissue-specific trait mean only in the individuals classified into the corresponding artificial tissue-specific subtype of BMI. Then for each trait, we computed central tendency measures of the artificial tissue-specific trait means across 500 sets of artificial tissue-specific eQTLs. For each trait, we also tested whether the overall mean of the artificial tissue-specific trait means is significantly different from the corresponding original (adipose or brain) tissue-specific trait mean. We performed the same experiment for WHRadjBMI.

**Permuting phenotype data across individuals.** Next, we performed a similar experiment for permuted phenotype (BMI or WHRadjBMI) data while keeping the eQTL assignment to tissues fixed as it is in the original data. We consider 500 random permutations of BMI across individuals. Keeping the genotype data fixed, we ran eGST for each permuted phenotype data and classified the tissue of interest across individuals based on 65% threshold of subtype posterior probability. As before, in each of these 500 pairs of subtype groups of individuals thus obtained, subtype-specific means were computed for each quantitative trait that was found to be primarily heterogeneous between the individuals of the original adipose (or brain) subtype of BMI and the remaining population (S10 Table). For each trait, we then computed central tendency measures of the tissue-specific means across 500 random BMI permutations. For each trait, we tested whether the overall mean of the tissue-specific trait means obtained across random permutations was significantly different from the corresponding original (adipose or brain) tissue-specific trait means. We conducted the same experiment for WHRadjBMI.

## Supporting information

**S1 Text. Markov Chain Monte Carlo (MCMC) algorithm for eGST.**
(PDF)

**S2 Text. Phenotypic characteristics of the tissue-specific groups of individuals for WHRadjBMI.**
(PDF)

**S1 Fig. Box plots of AUCs comparing the classification accuracy of eGST and the gold-standard strategy implementing our model in first set of simulation scenarios.**
(PDF)

**S2 Fig. Box plots of AUCs comparing the classification accuracy of eGST and the gold-standard strategy implementing our model in second set of simulation scenarios.**
(PDF)

**S3 Fig. Comparison between the true discovery rate of classifying tissue-specific subtypes by the MAP-EM algorithm versus the EM algorithm in first set of simulation scenarios.**
(PDF)

**S4 Fig. Comparison between the true discovery rate of classifying tissue-specific subtypes by the MAP-EM algorithm versus the EM algorithm in second set of simulation scenarios.**
(PDF)

**S5 Fig. Percentage of tissue-specific relative change of the quantitative traits differentially distributed between the individuals assigned to a tissue-specific subtype of WHRadjBMI and the remaining population.**
(PDF)

**S6 Fig. Percentage of tissue-specific relative change in the risk of case-control traits between the individuals assigned to a tissue-specific subtype of WHRadjBMI and the population.**
(PDF)

**S1 Table. Number of individuals classified by eGST as a tissue-specific subtype of BMI and WHRadjBMI.**
(PDF)

**S2 Table. Simulation results: Effect of increasing sample size on classification accuracy of eGST.**
(PDF)

**S3 Table. Simulation results: Effect of increasing number of tissue-specific SNPs with fixed subtype heritability per tissue on classification accuracy of eGST.**
(PDF)

**S4 Table. Simulation results: Effect of difference between baseline tissue-specific means of the phenotype ($\alpha_1$, $\alpha_2$) on the classification accuracy of eGST.**
(PDF)

**S5 Table. Simulation results: Effect of difference between the mean of tissue-specific genetic effect size distribution on the classification accuracy of eGST.**
(PDF)

**S6 Table. Pattern of first tissue-specific subtype posterior probability when a group of individuals have genetic effect from both tissues.**
(PDF)

**S7 Table. Simulation results: Percentage of misclassification when one of the two tissues is completely irrelevant to the phenotype.**
(PDF)

**S8 Table. Number of individuals in consecutive non-overlapping bins of BMI who were assigned to adipose and brain specific subtype of BMI, and the number of individuals that remained unassigned by eGST.**
(PDF)

**S9 Table. Number of individuals in consecutive non-overlapping bins of WHRadjBMI who were assigned to adipose and muscle specific subtype of WHRadjBMI, and the number of individuals that remained unassigned by eGST.**
(PDF)

**S10 Table. Quantitative traits among 106 phenotypes in UK Biobank which are differentially distributed between the adipose (and/or brain) specific subtype group of individuals for BMI and the remaining population.**
(PDF)

**S11 Table. Quantitative traits among 106 phenotypes in UK Biobank which are differentially distributed between the adipose subcutaneous (AS) (and/or muscle skeletal (MS)) specific subtype group of individuals for WHRadjBMI and the remaining population.**
(PDF)

**S12 Table. Case-control traits among 106 phenotypes considered in the UK biobank which are differentially distributed between the adipose (and/or brain) specific subtype group of individuals for BMI and the remaining population.**
(PDF)

**S13 Table. Case-control traits among 106 phenotypes considered in the UK biobank which are differentially distributed between the adipose subcutaneous (AS) (and/or muscle skeletal (MS)) specific subtype group of individuals for WHRadjBMI and the remaining population.**
(PDF)

**S14 Table. Qualitative/categorical traits with three or more categories that are differentially distributed between at least one of adipose subcutaneous (AS) and muscle skeletal (MS) specific subtype groups of individuals for WHRadjBMI and the remaining population.**
(PDF)

**S15 Table. Heterogeneity of non-BMI quantitative traits between the adipose (or brain) specific subtype group of individuals for BMI and the remaining population after BMI adjustment of the traits in the population using linear regression.**
(PDF)

**S16 Table. Phenotypic heterogeneity of non-WHR quantitative traits between adipose subcutaneous (AS) (muscle skeletal (MS)) specific group of individuals for WHRadjBMI and the remaining population after WHRadjBMI adjustment of the traits in the population using linear regression.**
(PDF)

**S17 Table. Magnitude of relative change of non-BMI quantitative traits between a tissue-specific subtype group of individuals for BMI and the corresponding group of BMI-matched individuals randomly selected from the population.**
(PDF)

**S18 Table. Magnitude of relative change of non-WHR quantitative traits between a tissue-specific subtype group of individuals for WHRadjBMI and the corresponding group of WHRadjBMI-matched individuals randomly selected from the population.**
(PDF)

**S19 Table. Summary of results from the analyses performed to characterize the tissue-specificity of phenotypic characteristics of individuals assigned to a tissue-specific subtype of BMI.**
(PDF)

**S20 Table. Summary of results from the analyses performed to characterize the tissue-specificity of phenotypic characteristics of the individuals assigned to a tissue-specific**

**subtype of WHRadjBMI.**
(PDF)

**S21 Table. Summary of results on the phenotypic characteristics of the individuals assigned to tissue-specific subtype of BMI identified based on random permutations of the primary phenotype (BMI) across individuals.**
(PDF)

**S22 Table. Summary of results on the phenotypic characteristics of the individuals assigned to tissue-specific subtype of WHRadjBMI identified based on random permutations of the primary phenotype (WHRadjBMI) across individuals.**
(PDF)

**S23 Table. Quantitative traits among 106 phenotypes in UK Biobank which are differentially distributed between the adipose-specific subtype group of individuals for BMI and brain-specific subtype group of individuals.**
(PDF)

## Acknowledgments

This research was conducted using the UK Biobank Resource under applications 24129 and 33297. We thank the participants of UK Biobank for making this work possible. We sincerely thank Malika Kumar Freund for her kind help with a part of the graphical presentation in the paper. We thank Huwenbo Shi, Kathryn Burch, Nicholas Mancuso, Hilary Finucane, Eran Halperin, Andy Dahl and Noah Zaitlen for helpful discussions relating to this work.

## Author Contributions

**Conceptualization:** Arunabha Majumdar, Bogdan Pasaniuc.

**Data curation:** Arunabha Majumdar, Na Cai.

**Formal analysis:** Arunabha Majumdar, Na Cai, Michael Gandal, Jonathan Flint.

**Funding acquisition:** Bogdan Pasaniuc.

**Investigation:** Arunabha Majumdar, Claudia Giambartolomei, Bogdan Pasaniuc.

**Methodology:** Arunabha Majumdar, Bogdan Pasaniuc.

**Resources:** Bogdan Pasaniuc.

**Software:** Arunabha Majumdar, Tanushree Haldar.

**Supervision:** Bogdan Pasaniuc.

**Validation:** Arunabha Majumdar, Bogdan Pasaniuc.

**Visualization:** Arunabha Majumdar, Claudia Giambartolomei, Tanushree Haldar.

**Writing – original draft:** Arunabha Majumdar.

**Writing – review & editing:** Arunabha Majumdar, Claudia Giambartolomei, Na Cai, Tommer Schwarz, Bogdan Pasaniuc.

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
