## [Decision Letter · Decision Letter 0]

24 Nov 2020

Dear Dr. Majumdar,

Thank you very much for submitting your manuscript "Leveraging eQTLs to identify individual-level tissue of interest for a complex trait" for consideration at PLOS Computational Biology. As with all papers reviewed by the journal, your manuscript was reviewed by members of the editorial board and by several independent reviewers. The reviewers appreciated the attention to an important topic. Based on the reviews, we are likely to accept this manuscript for publication, providing that you modify the manuscript according to the review recommendations.

Sincerely,

Ferhat Ay, Ph.D

Associate Editor

PLOS Computational Biology

Jian Ma

Deputy Editor

PLOS Computational Biology

[LINK]

Reviewer's Responses to Questions

**Comments to the Authors:**

Reviewer #1: In this manuscript, Majumdar et al. propose a novel method to quantify the tissue-wise genetic contribution to the trait of interest at the individual level. Briefly, they use a mixture model, which integrates tissue-specific eQTLs with genetic association data, to identify subgroups of individuals whose genetic predisposition act primarily through one specific trait. They showcase the utility of their method by extensive simulations and real data analyses of UK Biobank data. I find the manuscript is very interesting and may further help us understand the etiology of complex traits. However, I have a few questions/comments that hopefully the authors can consider to address.

1. A 65% threshold of tissue-specific subtype posterior probability was used. Is there any justification for this threshold? Are the results (especially those in real data analyses) sensitive to the choice of threshold?

2. How the tissue-specific genes and eQTL are selected? What's the specific cutoff we are using (on Line 583)? I just curious why "we included the top eQTL of the gene in both tissues, one in subcutaneous and one in visceral." Is there any consideration for that? Why not just simply exclude those genes to make the eQTL lists more tissue-specific?

Minor:

1. On Line 121, does the sigma_{y_k}^{2} is the same no matter C_i = 1 or C_i = 2? The trait variance seems the same as we deal with the same trait.

2. On line 468, some explanations regarding why using 5% are appreciated.

3. One line 432, I feel that starting with Model 2 may much easier to follow and understand. Model (1) seems misleading, especially if we read the Results section first. It is just a minor suggestion. It's Okay to keep the current form that starts with a more general model.

Reviewer #2: In this paper, the authors describe a statistical framework to infer the causal tissue per-individual underlying the predisposition for a specific complex trait. One of the application of this is to determine subtypes of a complex trait and assign individuals to it. While the aim of the study and the proposed methods are of interest, I do however have some comments/questions/concerns:

1. I would be curious to see what happens when using at least one irrelevant tissue (determined from biological knowledge or common sense). All the work described here starts from tissues that were previously inferred as relevant to the studied complex trait. What happens to the classification when one of the tissue being used is known to be independent of the trait? Are there any individual being assigned to this dummy tissue?

2. I understand that using genes specifically expressed for each tissue helps to avoid any ambiguities. However, GTEx introduced multiple methods and metrics to pinpoint causal eQTL variants and therefore accurately determine that a eQLT-eGene pair is specific to a given tissue. Can this information be used to improve the assignment (proportion & accuracy) of individuals to subtypes? If yes, to which extent?

3. In UK Biobank, >100,000 samples are related at least at the 3rd degree. How does this high level of relatedness between GWAS samples does affect your modeling? I understand that population stratification is accounted for thanks to PCs, but what about relatedness?

4. For BMI and WHRadjBMI, you managed to assign a tissue to 7.5% and 5.7% individuals, respectively. I understand that some of the causes for these relatively low percentages are technical, however is there also any biological rationale behind these?

5. When permuting the phenotype data in the case of real data, you get 7,404 and 3,433 individuals being assigned a tissue. On non-permuted data, you get 25,192 and 19,041 individuals with a tissue being assigned. Am I correct to say that you have 30% and 18% false discovery rate (FDR), respectively? If yes, can you comment on this?

6. When looking at the “phenotypic characteristics of individuals with an assigned tissue”, it seems to me that you mostly look at heterogeneity compared to general population. Could you actually compare individuals assigned to tissue A with those assigned to tissue B in a more systematic way and see if there is any phenotype significantly different? To me, that would be very informative to determine somehow a signature for the trait subtypes.

Reviewer #3: This study develops a methodology to quantify the tissue-specific genetic contribution to a trait for an individual. The Bayesian methodology uses tissue-specific eQTLs of tissue-specific genes to prioritize tissues. This approach can therefore be used to identify individuals for which the genetic contribution is mediated through the tissue (and therefore potentially identify disease subtypes). The authors then applied the methodology to BMI and WHRAdjBMI in the UK Biobank. There are major concerns with the approach which should be addressed, but I would recommend publication of the paper if these are adequately addressed because the study contains some interesting and novel insights.

One of the difficulties I have with the approach is that it does not explicitly consider the scenario in which the genetic contribution to the trait for an individual is mediated through multiple tissues. This scenario may well be the most generic one. The only model considered here is one in which the genetic contribution is mediated through a single tissue for an individual, and the extent to which this is realistic or plausible is not clear. The authors recognize this challenge, and, as they point out, in this case the approach would likely distribute the posterior probabilities equally across the tissues.

Practically, how would one choose the tissues to include in a general application? The authors focused on brain and adipose for BMI, but it's not clear how one would go about selecting tissues to include for a different phenotype.

Also, the authors used only the top eQTL for each tissue-specific gene. What's the distribution of the number of independent eQTLs for the tissue-specific genes? This will help to clarify/quantify the limitation of the approach.

It's not clear how the uncertainty in the input set of "tissue-specific genes" or the set of tissue-specific eQTLs for such a gene affects the quantification (posterior probability) of the mediating tissue.

The authors refer to an R package. Is the source available through github or some other repository? The link should be included.

**Have all data underlying the figures and results presented in the manuscript been provided?**

Reviewer #1: Yes

Reviewer #2: Yes

Reviewer #3: **No: **Please include the link to the source code.

PLOS authors have the option to publish the peer review history of their article (what does this mean?). If published, this will include your full peer review and any attached files.

Reviewer #1: **Yes: **Chong Wu

Reviewer #2: No

Reviewer #3: No
---

## [Decision Letter · Decision Letter 1]

26 Mar 2021

Dear Dr. Majumdar,

We are pleased to inform you that your manuscript 'Leveraging eQTLs to identify individual-level tissue of interest for a complex trait' has been provisionally accepted for publication in PLOS Computational Biology.

Best regards,

Ferhat Ay, Ph.D

Associate Editor

PLOS Computational Biology

Jian Ma

Deputy Editor

PLOS Computational Biology

Reviewer's Responses to Questions

**Comments to the Authors:**

Reviewer #1: The authors have addressed all my comments.

Reviewer #3: The authors have fully addressed my concerns. The study, as I noted previously, contains some interesting insights, and so I recommend publication.

**Have all data underlying the figures and results presented in the manuscript been provided?**

Reviewer #1: Yes

Reviewer #3: Yes

PLOS authors have the option to publish the peer review history of their article (what does this mean?). If published, this will include your full peer review and any attached files.

Reviewer #1: No

Reviewer #3: **Yes: **Eric R. Gamazon

---

## [Editor Report · Acceptance letter]

17 May 2021

PCOMPBIOL-D-20-01662R1 

Leveraging eQTLs to identify individual-level tissue of interest for a complex trait

Dear Dr Majumdar,

I am pleased to inform you that your manuscript has been formally accepted for publication in PLOS Computational Biology. Your manuscript is now with our production department and you will be notified of the publication date in due course.

With kind regards,

Zita Barta
